# HIV-1 Gag targeting to the plasma membrane reorganizes sphingomyelin-rich and cholesterol-rich lipid domains

Nario Tomishige [1,2] ✉, Maaz Bin Nasim[1,3], Motohide Murate [1,2], Brigitte Pollet[1], Pascal Didier[1], Julien Godet[1], Ludovic Richert [1], Yasushi Sako [2], Yves Mély [1] ✉ & Toshihide Kobayashi [1,2] ✉

Although the human immunodeficiency virus type 1 lipid envelope has been reported to be enriched with host cell sphingomyelin and cholesterol, the molecular mechanism of the enrichment is not well understood. Viral Gag protein plays a central role in virus budding. Here, we report the interaction between Gag and host cell lipids using different quantitative and super-resolution microscopy techniques in combination with specific probes that bind endogenous sphingomyelin and cholesterol. Our results indicate that Gag in the inner leaflet of the plasma membrane colocalizes with the outer leaflet sphingomyelin-rich domains and cholesterol-rich domains, enlarges sphingomyelin-rich domains, and strongly restricts the mobility of sphingomyelin-rich domains. Moreover, Gag multimerization induces sphingomyelin-rich and cholesterol-rich lipid domains to be in close proximity in a curvature-dependent manner. Our study suggests that Gag binds, coalesces, and reorganizes pre-existing lipid domains during assembly.

Human immunodeficiency virus type 1 (HIV-1) lipid envelope is obtained during budding from the plasma membrane (PM) of infected host cells. Various lipidomics studies indicate that the lipid composition of the viral membrane differs from that of the producer cell. Virus particles are enriched in sphingomyelin (SM), cholesterol (Chol), ganglioside GM3, and phosphatidylinositol diphosphate (PIP$_2$) whereas they show reduced levels of phosphatidylinositol and unsaturated phosphatidylcholine (PC) species[1–5]. These results suggest that the virus buds from specific lipid domains of the PM or the virus induces the formation of specific lipid domains. Accumulating evidence indicates the importance of the lipid composition of the HIV-1 membrane during virus entry and budding[6–10].

Although the specific lipid composition of the HIV-1 envelope is well recognized, the molecular mechanisms of the selection of specific lipids from the host cell are not well understood. The minimal component required for HIV-1 assembly at the PM is the viral Gag protein since its expression is sufficient to promote the formation of virus-like

particles carrying a lipidic envelope derived from the host cell membrane[11,12]. Gag is synthesized in the cytosol as a 55 kDa polyprotein comprising several domains that are cleaved into independent proteins after budding[13]. The binding of Gag to genomic RNA in the cytoplasm is accompanied by oligomerization of Gag[14–17]. Gag oligomers are then targeted to the site of budding where they interact with the membrane and further multimerize[18]. The targeting of Gag to the PM is dependent on negatively charged lipids, especially phosphatidylinositol 4,5-bisphosphate (PI(4,5)P$_2$)[9,19–21] as well as on Chol[22,23].

In the PM of mammalian cells, lipids are asymmetrically distributed: PI(4,5)P$_2$, phosphatidylethanolamine, and phosphatidylserine are in the inner leaflet whereas PC, SM, and glycolipids are mainly located in the outer leaflet[24–26]. One major question in the assembly of HIV-1 is how the binding of Gag to inner leaflet PI(4,5)P$_2$ recruits SM and glycosphingolipids in the outer leaflet. Colocalization of Gag and ganglioside GM1 labeled with cholera toxin has been shown by fluorescence microscopy[27,28]. However, lipid domains are heterogeneous[29],

[1]Laboratoire de Bioimagerie et Pathologies, UMR 7021 CNRS, Faculté de Pharmacie, Université de Strasbourg, Illkirch, France. [2]Cellular Informatics Laboratory, RIKEN CPR, Wako, Saitama, Japan. [3]Present address: Faculty of Pharmacy, The University of Lahore, Lahore, Pakistan. ✉e-mail: nario.tomishige@unistra.fr; yves.mely@unistra.fr; toshihide.kobayashi@unistra.fr

and SM-rich domains have been shown to segregate from GM1-rich domains by electron microscopy[30]. In addition, GM1 has been shown to be trapped only transiently in viral assembly sites[28].

Together with Chol, SM, and glycolipids are postulated to form specific lipid domains, whose estimated diameter is around 5–50 nm[31–35]. Since the diameter of an HIV-1 particle is 100–150 nm[36–39], the area of the plasma membrane needed for its formation is about 200–300 nm in diameter. Thus, it is unlikely that a virus particle assembles within and buds from a single lipid domain. Rather, it is more likely that virus particle assembly involves the recruitment and coalescence of small lipid domains at the assembly sites[7]. In line with this hypothesis, Gag has been reported to induce coalescence of lipid raft domains and tetraspanin-enriched domains[40]. However, little is known about how Gag reorganizes lipid domains. Recently Sengupta et al. reported the co-localization of fluorescent SM and Chol analogs with expressed Gag proteins on the plasma membrane[41]. Moreover, using different fluorescent analogs, Favard et al. showed that the surface expression of Gag restricted the mobility of Chol but not that of a SM analog[42]. However, the physical properties of fluorescent lipid analogs significantly differ from those of natural counterparts and can thus bias the conclusions[43,44]. To our knowledge, the effect of Gag on the distribution and dynamics of endogenous host lipids has not been examined.

In this context, our aim in the present study was to examine the interaction of Gag with endogenous SM and Chol in the PM of Gag-transfected HeLa cells using different optical microscopy techniques in combination with lipid-specific probes. We have developed and/or characterized various proteins that bind specific lipids, including non-toxic lysenin (NT-Lys), a SM-binding protein that does not crosslink SM[30] and D4, a Chol-specific probe[45,46]. Our results indicate transbilayer co-localization of Gag and SM-rich domains, Gag-induced restriction of the mobility of endogenous SM-rich domains, and reorganization of SM-rich and Chol-rich domains in a Gag-oligomerization- and curvature-dependent manner.

## Results

An enrichment of SM and Chol in the HIV-1 envelope has been described[1,3,5]. Although Chol has a higher affinity to SM over other phospholipids[47–49], its relative abundance (40% of the PM lipids) compared to SM (10–15%), suggests that Chol also interacts with other PM lipids. In this study, we examined the effect of Gag expression on the cell surface distribution and dynamics of endogenous SM-rich and Chol-rich domains. To visualize these lipid domains, we used N-terminally truncated lysenin (NT-Lys)[30] and the D4 domain of perfringolysin O (PFO)[45]. Lysenin is an earthworm-derived protein toxin[50,51] that specifically binds SM[52] and oligomerizes to form a nonameric pore[53–56]. Lysenin selectively binds to clusters of 5–6 molecules of SM[57]. NT-Lys is a truncated mutant of lysenin deficient in its N-terminal 160 amino acids that are required for oligomerization and protein toxicity[30]. Similar to lysenin, NT-Lys binds SM clusters but does not oligomerize nor induce lipid clustering[30,58]. NT-Lys is very slowly endocytosed[30], indicating that NT-Lys selectively labels cell surface SM-rich lipid domains. D4 domain of PFO is a non-toxic Chol-binding fragment of the Chol-binding pore-forming toxin, PFO[45], and binds Chol when Chol concentration in the membrane is higher than 40%[59,60]. NT-Lys and D4 domain keep their specificity after labeling with fluorescent dyes Alexa Fluor 647 and Janelia Fluor 549[61] via SNAP-tag (AF647-NT-Lys, AF647-D4, and JF549-D4, Supplementary Fig. 1).

To observe the localization of Gag molecules on the PM and their effects on lipid organization, we used fluorescent Gag constructs. Previously, expression of Gag-GFP was shown to lead to assembly/budding defects when GFP was conjugated to the C-terminus of Gag[62]. In contrast, when EGFP was inserted between MA and CA domains of Gag, the transfection of Gag-EGFP: non-tagged Gag (1:1) produced comparable amounts of viral particles to the transfection of non-

tagged Gag alone[63]. Therefore, in this study, we inserted FPs between the MA and CA domains of Gag[16]. To limit the steric hindrance of FP on Gag assembly, we used a ratio of 3:1 for non-tagged and FP-tagged Gag plasmid to transfect cells throughout the experiments. We confirmed that the production and budding of virus-like particles (VLPs) from cells transfected with Gag/Gag-FP at this ratio was not affected as monitored by Western blotting against anti-p24 Gag antibody and electron microscopy (Supplementary Figs. 2–4).

### Expression of Gag does not alter the gross lipid composition of HeLa cells

The labeling of SM-rich lipid domains by NT-Lys depends on the content of SM[58], Chol[46], and glycolipids[57] of the PM. Similarly, the labeling of Chol-rich domains by D4 depends on the availability of Chol for the binding that is determined by the lipid composition in the environment[64,65]. At the beginning of this study, we, therefore, examined the possibility that Gag expression causes an alteration in lipid composition by investigating the contents of polar and neutral lipids in control cells and cells expressing Gag and Gag mutants (Fig. 1a, c and Supplementary Figs. 4 and 5). Figure 1b, d show that SM and Chol contents normalized by protein contents were not significantly different among these cells. In addition, the contents of other lipids were also not significantly affected (Fig. 1a, c). These results indicate that Gag and Gag mutant expression did not significantly affect the gross lipid composition of HeLa cells.

### Transbilayer co-localization of Gag and endogenous SM-rich and Chol-rich lipid domains in the PM

Next, we examined the effect of Gag and Gag derivatives, expressed in the cytoplasm, on the distribution and dynamics of endogenous SM- and Chol-rich lipid domains in the outer leaflet of the plasma membrane where the two lipids can interact with each other. The binding of Gag to the PM is dependent on PI(4,5)P$_2$[9,19–21] and Chol[22,23]. Previously we showed that cell surface SM labeled with NT-Lys[46] and SM/Chol complexes labeled with a protein specific for this complex of lipids, nakanori, partially colocalized with inner leaflet PI(4,5)P$_2$ labeled by PH domain[32]. Here we first examined the effect of Gag assembly on transbilayer co-localization of Gag and the above two lipid domains. To this end, we compared wild-type Gag protein (Gag WT) and Gag-WM mutant that harbors two mutations (W184A and M185A) positioned in helix 9 of the CA-dimer interface[66–68]. This Gag mutant traffics and binds to the inner leaflet of the PM, but is defective in multimerization[9,69,70] and fails to induce the clustering of PI(4,5)P2 in liposomes[9,70]. HeLa cells were transfected with a mixture of plasmids encoding Gag WT and Gag WT-mTagBFP2, or Gag-WM and Gag-WM-mTagBFP2. After 24 h, the cells were labeled with mCherry-D4 and EGFP-NT-Lys by adding the lipid-binding probes to the medium. Under a confocal microscope, both lipid probes labeled the entire cell surface, but somehow non-uniformly in the case of mCherry-D4. Compared to the WM mutant, Gag WT-mTagBFP2 tended to be unevenly distributed at the PM, but both Gag proteins colocalized well with lipid probes (Fig. 2a). However, this conclusion may be biased by the diffraction-limited resolution of confocal microscopy since previous electron microscopy and super-resolution microscopy studies showed that NT-Lys and lysenin labeled SM-rich lipid domains were of 20–250 nm in diameter[26,30,32,46,71], and D4 labeled Chol-rich lipid domains were of around 120 nm in radius[71]. Therefore, we next investigated at higher resolution the localization of Gag and NT-Lys labeled SM-rich or D4-labeled Chol-rich domains using photoactivation localization microscopy/direct stochastic optical reconstruction microscopy (PALM/dSTORM). To this end, we transfected HeLa cells with Gag/Gag-mEos4b or Gag-WM/Gag-WM-mEos4b (Supplementary Figs. 4 and 6) and labeled the cells with AF647-NT-Lys or AF647-D4 after 24 h of transfection. We then recorded fluorescence images followed by

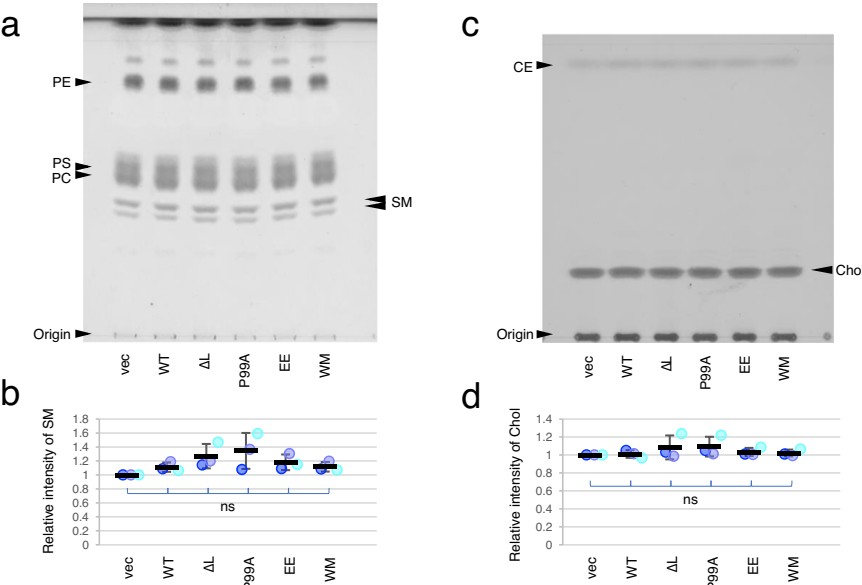

**Fig. 1 | Lipid composition was not affected by Gag and Gag derivatives expression.** Total lipid fraction extracted from HeLa cells transfected with vectors or plasmids encoding the indicated Gag and Gag derivatives was separated in a solvent for polar lipids (**a**) or neutral lipids (**c**) as described in "Methods". The bands corresponding to SM (**a**) and Chol (**c**) indicated by arrowheads on the right side of thin layer chromatography (TLC) plates were quantitated. The relative band intensity to the control (vectors) was shown as a mean ± SD for three independent experiments in (**b**) and (**d**). The values in each experiment were also shown in dots in three different blue. The differences were not significant (ns) among all samples at the significance level of 0.05 in one-way ANOVA post-hoc Tukey test. Band positions of phosphatidylcholine (PC), phosphatidylserine (PS), phosphatidylethanolamine (PE), and cholesteryl ester (CE) were indicated by arrowheads on the left side of TLC plates in (**a**) and (**c**). Vec, vectors; WT, Gag WT; ΔL, Gag-ΔL; P99A, Gag-P99A; EE, Gag-EE; WM, Gag-WM. Source data and statistics for (**b** and **d**) were provided as a Source Data file.

the analysis of 4–6 cell images per sample in each of three independent experiments.

PALM/dSTORM visualized domains of different sizes labeled with the two fluorophores (Fig. 2b, g, Supplementary Figs. 7 and 8). Although Gag WT-mEos4b did not completely overlap with AF647-NT-Lys or AF647-D4, it was often associated with AF647-NT-Lys or AF647-D4 domains (yellow dotted lines in Fig. 2c, h), in line with the confocal microscopy data. Gag-WM-mEos4b was either associated with AF647-NT-Lys or localized in areas where AF647-NT-Lys or AF647-D4 signal was sparse (Fig. 2d, i, yellow arrows). To analyze the colocalization between Gag-mEos4b and AF647-NT-Lys or AF647-D4, we calculated the Mander's overlap coefficient (MOC) using Coloc-Tesseler[72], which is based on overlapped Voronoi diagrams of the two molecular species and also implemented in PoCA program[73] A Voronoi diagram is a tessellation where a tile corresponding to a given data point is a locus of all points in space closest to this data point[74]. Figure 2e, f shows that 55% of Gag WT-mEos4b colocalized with 26% of AF647-NT-Lys. The difference in the two values may reflect the abundance of SM-rich domains compared to Gag in the PM. In contrast to Gag WT, only 27% of Gag-WM-mEos4b colocalized with 12% of AF647-NT-Lys (Fig. 2e, f). In the colocalization analysis of Gag-mEos4b and AF647-D4, 52% of Gag WT-mEos4b colocalized with 27% of AF647-D4 whereas only 25% of Gag-WM-mEos4b colocalized with 25% of AF647-D4 (Fig. 2j, k). These results indicated that Gag WT-mEos4b shows the same level of transbilayer colocalization with AF647-NT-Lys-labeled SM-rich and AF647-D4-labeled Chol-rich domains and that this colocalization between Gag and lipid domains is dependent on Gag multimerization, as shown by the comparison of MOC between Gag WT and Gag-WM.

### Gag association enlarges AF647-NT-Lys-positive lipid domains

We next examined the effect of Gag expression on the organization of SM-rich and Chol-rich lipid domains. To compare the size of these domains in the absence and presence of Gag expression, we estimated the sizes of these domains and the localization numbers within these domains by SR-Tesseler[75], a program based on Voronoi tessellation of individual localizations and also implemented in the PoCA program (see "Methods"). Gag WT-mEos4b formed a round shape of domains isolated from other domains and large domains including several round domains as shown by arrows and arrowheads, respectively in Supplementary Fig. 9. We analyzed all AF647-NT-Lys and Gag-mEos4b domains including 10 or more localizations that were detected in a whole cell area of 4–6 cell images per sample in three independent experiments (Fig. 3). The mean (±SEM) diameter of AF647-NT-Lys domains (WT, 220 ± 10 nm) was found to be significantly larger in cells expressing Gag WT/Gag WT-mEos4b than in cells without Gag expression (vec, 170 ± 10 nm) or cells expressing Gag-WM/Gag-WM-mEos4b (WM, 180 ± 20 nm) (Fig. 3a). When examining individual domains, a decrease in the population of AF647-NT-Lys domains smaller than 128 nm was observed in cells expressing Gag WT compared to vector control (Supplementary Fig. 10a and Supplementary Table 1). This reduction seemed to be compensated by a significant increase in the population of domains larger than 426 nm. The WM mutant-expressing cells showed a distribution of the domain diameters similar to the vector control. The domain density of AF647-NT-Lys that was calculated by dividing the number of AF647-NT-Lys localizations included in each domain by the domain area decreased in the presence of Gag WT compared to that in the absence of Gag (Fig. 3b, Supplementary Fig. 10b, c, and Supplementary Table 2). The mean diameters of Gag WT-mEos4b domains (180 ± 10 nm, mean ± SEM) were greater than that of Gag-WM-mEos4b domains (119 ± 9 nm) (Fig. 3c).

In contrast to NT-Lys, the mean diameters of AF647-D4 domains were around 170 nm and not significantly different among samples without and with Gag and with Gag-WM expression (Fig. 3d). These mean diameters of AF647-D4 domains were in the same range as the diameter of AF647-NT-Lys domains in the absence of Gag. When examining individual AF647-D4 domains, a population in the area less

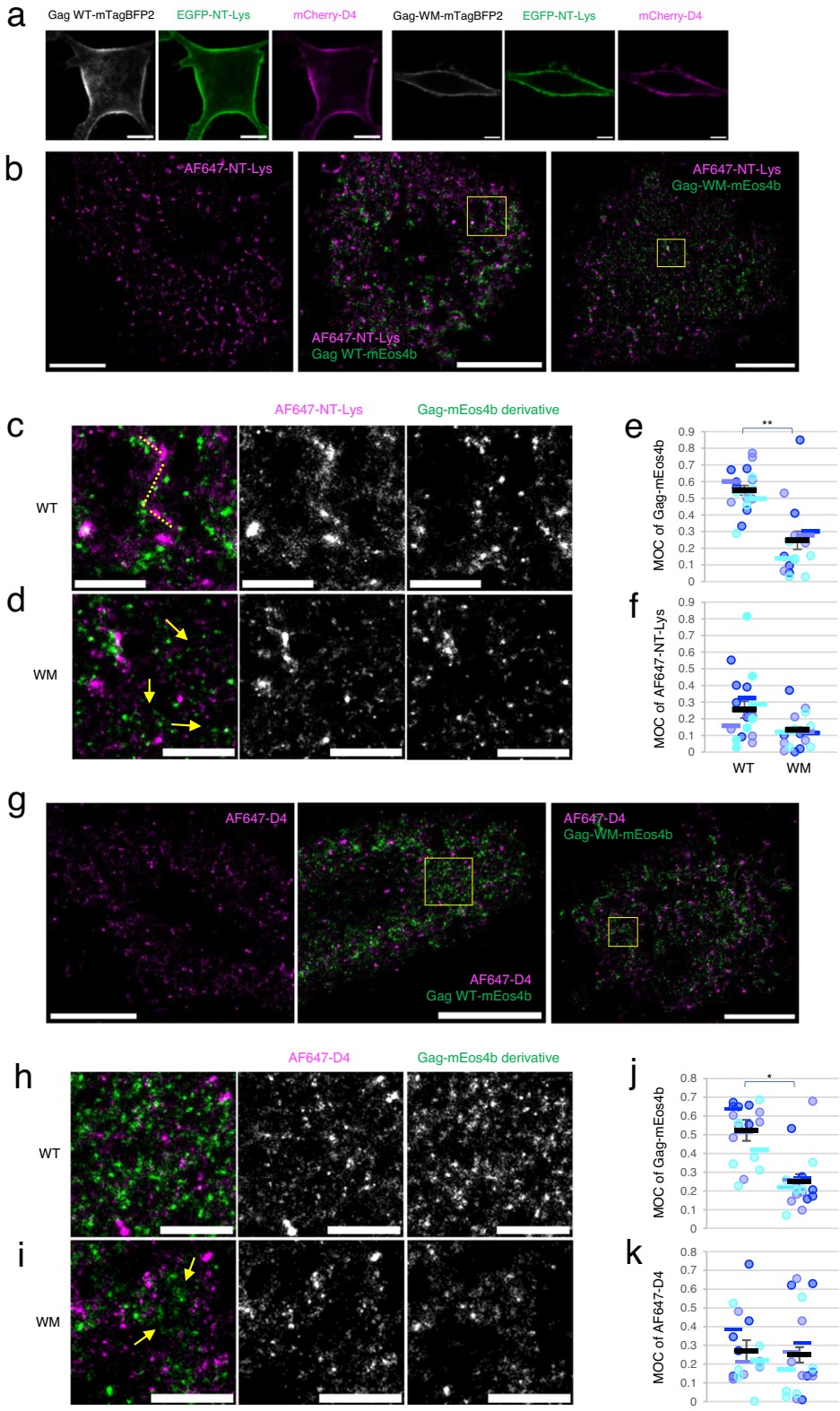

than 2000 nm² decreased upon Gag WT expression (Supplementary Fig. 11a and Supplementary Table 4) and the number of AF647-D4 localizations per domain did not change (Supplementary Fig. 11b and Supplementary Table 5), resulting in a decreased density of AF647-D4 domains in the presence of Gag WT (Fig. 3e, Supplementary Fig. 11c and Supplementary Table 6). Domain diameters of Gag-mEos4b

derivatives were similar for cells labeled with AF647-NT-Lys and AF647-D4 (Fig. 3c, f), indicating that the labeling by lipid probes did not impact Gag localization.

These results indicated that Gag expression induced the formation of larger SM-rich domains in a multimerization-dependent manner.

**Fig. 2 | Colocalization between Gag-mEos4b and AF647-NT-Lys in PALM/dSTORM imaging. a** Confocal images of Gag/Gag-mTagBFP2 derivatives-expressing HeLa cells labeled with EGFP-NT-Lys and mCherry-D4. Bar, 10 μm. **b, g** PALM/dSTORM images of Gag/Gag-mEos4b derivatives-expressing cells labeled with AF647-NT-Lys (**b**) and AF647-D4 (**g**). Bar, 5 μm **c, d, h,** and **i** Regions enclosed by yellow rectangles in (**b**) and (**g**) are enlarged. **c** Images in a cell expressing Gag WT/Gag WT-mEos4b and labeled with AF647-NT-Lys. The yellow dotted lines indicate a similar pattern of labeling with both fluorophores. **d** Images in a cell expressing Gag-WM/Gag-WM-mEos4b and labeled with AF647-NT-Lys. The yellow arrows indicate Gag-WM-mEos4b signals that were not associated with AF647-NT-Lys signals (see text). **h** Images in a cell expressing Gag WT/Gag WT-mEos4b and labeled with AF647-D4. **i** Images in a cell expressing Gag-WM/Gag-WM-

mEos4b and labeled with AF647-D4. The yellow arrows indicate Gag-WM-mEos4b signals that looked independent of AF647-D4 signals. Bar, 1 μm. **e, f** Mander's overlap coefficient (MOC) values for colocalization of Gag WT-mEos4b (WT) and Gag-WM-mEos4b (WM) to AF647-NT-Lys (**e**) and of AF647-NT-Lys to WT and WM (**f**). **j, k** MOCs of WT and WM to AF647-D4 (**j**) and of AF647-D4 to WT and WM (**k**). In the graphs, dots and bars in three different blue and black bars with error bars indicate means in each cell, means in each experiment, and means ± SEM in three independent experiments (n = 18 for WT and 16 for WM over three experiments). * and ** in the graphs indicate significant differences at significance levels of 0.05 and 0.01 in the paired two-tailed t-test. Source data and p-values for (**e, f, j,** and **k**) were provided as a Source Data file.

## Expression of Gag restricts the lateral diffusion of cell surface SM-rich domain

The observed transbilayer co-localization of Gag in the inner leaflet and NT-Lys bound to SM-rich domain or D4 to Chol-rich domain in the outer leaflet of the PM (Fig. 2) suggested an interaction between Gag in the inner leaflet and SM and/or Chol in the outer leaflet. Moreover, Gag was shown to enlarge lipid domains labeled with AF647-NT-Lys (Fig. 3 and Supplementary Fig. 10). We then interrogated whether Gag alters the dynamics of endogenous SM and Chol domains. To this end, we measured fluorescence recovery after photobleaching (FRAP) of EGFP-NT-Lys and EGFP-D4 in the absence and presence of Gag expression. HeLa cells were transfected either with empty vectors or a mixture of Gag WT and Gag WT-mCherry. After 22 h, at which time most trans-fected cells exhibited Gag-mCherry fluorescence at the PM (Fig. 4a, e), we measured FRAP in cells labeled with EGFP-NT-Lys or EGFP-D4 in each condition and then drew recovery curves (Fig. 4b, c, f, and g). The average diffusion coefficients of EGFP-NT-Lys were marginally decreased by Gag expression (Fig. 4d; control cells, mean ± SD = 0.013 ± 0.002 μm²/s; Gag transfected cells, 0.010 ± 0.002 μm²/s). In contrast, at the end of the chase, fluorescence recoveries of EGFP-NT-Lys indicated that the immobile fraction of EGFP-NT-Lys molecules was increased from 15% to 36% by Gag expression. This significant increase in the immobile fraction suggested that Gag at the inner leaflet of the PM strongly restricted the diffusion of the SM-rich lipid domains at the outer leaflet. As the diffusion constant of the mobile fraction was not significantly affected by Gag, this mobile fraction may correspond to SM-rich lipid domains not in transbilayer contact with Gag proteins. In the case of EGFP-D4, interestingly, there was a large portion (80%) of immobile fraction irrespective of Gag expression. The differences in the mobile fraction as well as the diffusion constant were not evident between the absence and presence of Gag (Fig. 4g, h).

These results suggest that Gag restricts the diffusion of SM-rich domains across the bilayer. On the other hand, Chol-rich domains labeled with EGFP-D4 are more immobile than SM-rich domains labeled with EGFP-NT-Lys even in the absence of Gag.

## Gag-dependent reorganization of SM-rich lipid domains detected by FLIM-FRET

The above results suggest that Gag on the cytoplasmic leaflet of the PM beneath SM-rich domains restricts the diffusion of SM-rich domains (Fig. 4) and enlarges the SM-rich domains (Fig. 3). We further investi-gated the formation of larger SM-rich domains by monitoring the Gag-induced changes in the intermolecular distances between the NT-Lys molecules bound to SM-rich domains. These intermolecular distances were analyzed by fluorescence lifetime imaging based on Förster resonance energy transfer (FLIM-FRET) measurements using EGFP-NT-Lys as a FRET donor and mCherry-NT-Lys as a FRET acceptor. Assum-ing that non-energy transferring EGFP-NT-Lys proteins (at distances >10 nm from the closest acceptor) coexist with transferring EGFP-NT-Lys proteins (at distances <10 nm from one or several close acceptors), two populations of lifetimes are expected ("Methods" and Supple-mentary Note). The non-transferring population is thought to exhibit a

lifetime, $\tau_2$, corresponding to EGFP-NT-Lys alone while the transferring one should exhibit a shorter lifetime, $\tau_1$. We thus analyzed the FLIM data in cells doubly labeled with donor and acceptor with a double exponential equation.

At 24 h post-transfection with vectors, Gag WT/Gag WT-mTagBFP2, or Gag-WM/Gag-WM-mTagBFP2, HeLa cells were labeled with EGFP-NT-Lys alone or doubly labeled with EGFP-NT-Lys and mCherry-NT-Lys (Supplementary Fig. 12). The data were obtained from 9 or 10 cells in each of three independent experiments.

First, we analyzed the mean lifetime by a single exponential equation. The cells labeled with EGFP-NT-Lys alone showed a mean lifetime (±SEM) of $2.2 ± 2.0 × 10^{-2}$ ns (Fig. 5a), which was close to the lifetime of free EGFP (2.4 ns)[76]. When cells were doubly labeled with EGFP-NT-Lys and mCherry-NT-Lys, the lifetime was dramatically decreased to a mean at $1.3 ± 8.0 × 10^{-2}$ ns, indicating the occurrence of FRET between donor and acceptor molecule (Fig. 5a). In Fig. 5b, c, we analyzed with a double exponential equation the effect of Gag WT/Gag WT-mTagBFP2 expression on FRET between EGFP-NT-Lys and mCherry-NT-Lys. In the absence of Gag, doubly labeled cells (DA) showed a mean lifetime $\tau_1$ of 0.96 ± 0.09 ns and a mean amplitude of 82 ± 1 %. This amplitude indicates that approx. 80% of the NT-Lys molecules are in close proximity to each other on the PM, in line with the PALM/dSTORM data (Figs. 2 and 3), confirming that EGFP-NT-Lys/mCherry-NT-Lys molecules accumulate in specific lipid domains. Expression of Gag decreased the lifetime $\tau_1$ from 0.96 ± 0.09 ns to 0.88 ± 0.09 ns and increased the $\alpha_1$ amplitude from 82 ± 1 % to 88.1 ± 0.9 % (Fig. 5b, c). A shortening of lifetime in FRET can result from a closer distance between a donor and an acceptor and/or the pre-sence of multiple acceptors around a donor[77–80], and an increase in its amplitude indicates an increase in the transferring population. These results, therefore, support the observation that Gag induces the for-mation of larger SM-rich lipid domains. We then investigated whether the formation of larger domains depends on Gag multimerization using WM mutant. Unexpectedly, the lifetime $\tau_1$ (0.91 ± 0.10 ns) and its amplitude $\alpha_1$ (87 ± 1 %) did not significantly differ from the WT samples and the lifetime did not significantly differ from the DA sample as well (Fig. 5b, c). This indicated that Gag WT reorganizes the SM-rich domains to form larger domains. The WM mutant also reorganizes the SM-rich lipid domains, but not as efficiently as Gag WT.

## Gag multimerization induces SM-rich and Chol-rich domains to be in close proximity

To better understand the possible effect of Gag on the lipid distribu-tion in the PM, we investigated by FLIM-FRET the interaction of Chol-rich lipid domains with SM-rich lipid domains and its dependence on Gag multimerization.

When cells were doubly labeled with EGFP-NT-Lys (FRET donor) and mCherry-D4 (FRET acceptor) in the absence of Gag, a single component analysis showed a reduction of the lifetime from $2.2 ± 4.0 × 10^{-4}$ (mean ± SEM) ns in cells labeled with EGFP-NT-Lys alone to $2.1 ± 1.0 × 10^{-2}$ ns, indicating FRET between the donor and acceptor (Fig. 6a). Next, we analyzed data in cells doubly labeled with

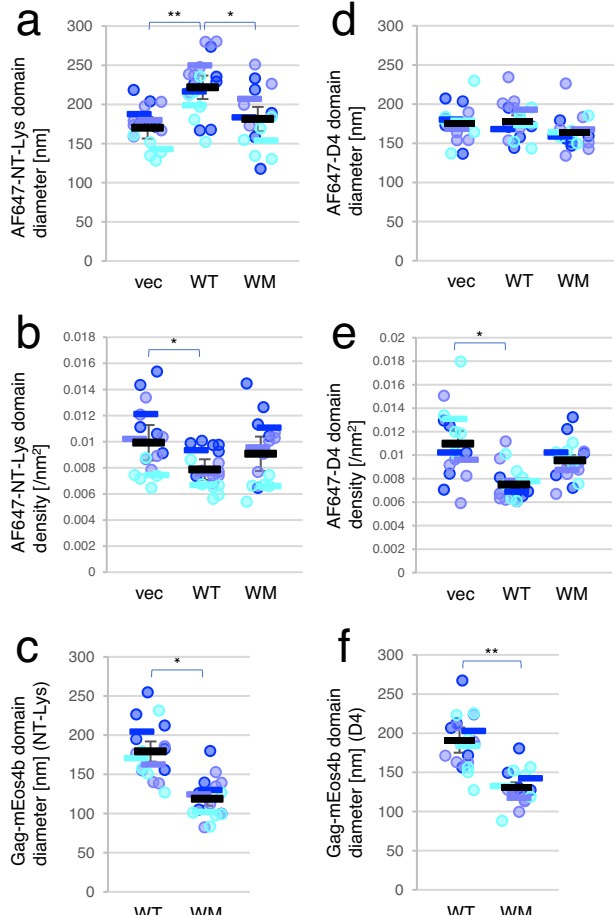

**Fig. 3 | Gag WT expression induces the formation of larger AF647-NT-Lys domains but not AF647-D4 domains.** The mean diameter [nm] (**a**) and mean density [/nm²] (**b**) of AF647-NT-Lys domains, and the mean diameter [nm] of Gag-mEos4b domains (**c**) in cells labeled with AF647-NT-Lys, and the mean diameter [nm] (**d**) and mean density [/nm²] (**e**) of AF647-D4 domains, and the mean diameter [nm] of Gag-mEos4b domains (**f**) in cells labeled with AF647-D4 in PALM/dSTORM. Vec, cells expressing empty vectors; WT, Gag WT; WM, Gag-WM. The mean value of domains in each cell, the means in each experiment, and the mean ± SEM of the three independent experiments were shown as dots and bars in three different blue colors and black bars with error bars ($n = 17$, 18, and 16 (**a**–**c**), and $n = 15$, 16, and 16 (**d**–**f**) for vec, WT, and WM over three experiments). * and ** indicate significant differences at the significance level of 0.05 and 0.01, respectively, in the one-way ANOVA post-hoc Tukey test (**a**, **b**, **d**, and **e**) and paired two-tailed $t$-test (**c** and **f**). Source data and detailed statistics were provided as a Source Data file.

donor and acceptor with the double exponential model. Means of lifetime $\tau_1$ and amplitude $\alpha_1$ were $1.4 \pm 4.0 \times 10^{-2}$ ns and $26.2 \pm 0.7$ %, respectively (Fig. 6b, c). In addition, to visualize the distribution of lifetime $\tau_1$ and amplitude $\alpha_1$ in the interacting population, we applied a recently devised FLIM diagram plot[79] to interacting populations in a representative experiment ($n = 10$ cells). The interacting species were distributed in two populations peaking at $\tau_1$, $\alpha_1$ of 1.9 ns, 30%, and 1.1 ns, 20%, respectively (DA in Supplementary Fig. 13). These lifetimes corresponded to FRET efficiencies of ~12% and 49%, respectively. These two populations with broad distributions suggest the existence of heterogeneous domains containing both SM and Chol on the surface of HeLa cells. As mentioned above, the lifetime reduction indicates that the population with $\tau_1 = 1.1$ ns may correspond to donor molecules that have shorter distances with acceptors and/or more acceptors around them than those in the population with $\tau_1 = 1.9$ ns.

We then examined whether Gag re-organizes the SM-rich and Chol-rich lipid domains. Expression of Gag WT marginally changed the mean lifetime $\tau_1$ (~1.4 ns) but significantly increased the mean amplitude $\alpha_1$ from $26.2 \pm 0.7$% to $30.2 \pm 0.9$% (Fig. 6b, c). The FLIM diagram plot in the presence of Gag WT revealed that $\tau_1$ and $\alpha_1$ values were still scattered in two populations peaking at 1.9 ns with an amplitude of 35% and at 1.2 ns with 25% (WT in Supplementary Fig. 13). Compared to cells that did not express Gag, the population peaking at 1.9 ns decreased and the population peaking at 1.1 to 1.2 ns increased from 20% to 25% upon Gag expression (DA and WT in Supplementary Fig. 13). These data suggest that Gag proteins induce a closer apposition between SM-rich and Chol-rich domains in the PM.

We next investigated the role of Gag multimerization on the FLIM-FRET between EGFP-NT-Lys and mCherry-D4. When the non-multimerizable Gag-WM mutant was expressed, the mean values in the lifetime $\tau_1$ and the amplitude $\alpha_1$ (Fig. 6b, c), as well as the FLIM diagram plot (WM and DA in Supplementary Fig. 13), were very similar to those without Gag.

In addition, careful observation of Gag-mTagBFP2 and lifetime images suggested a correlation between Gag-mTagBFP2 intensity and shortening in the lifetime of EGFP-NT-Lys in cells expressing Gag WT but not Gag-WM (Fig. 6d and Supplementary Fig. 14). To quantitate this correlation, we plotted $\tau_1$ and $\alpha_1$ in every quartile of Gag-mTagBFP2 intensity and calculated Pearson correlation coefficients between mTagBFP2 intensities and means of either $\tau_1$ or $\alpha_1$ values (Fig. 6e, f, and "Methods"). Consistent with the observation in Fig. 6d, there were negative or positive correlations between mTagBFP2 intensities and $\tau_1$ or $\alpha_1$ values, respectively, in Gag WT-expressing cells but not in Gag-WM-expressing cells (Fig. 6e, f). This indicates that the reorganization of SM-rich and Chol-rich domains is related to Gag multimerization.

We confirmed the above FLIM-FRET results with super-resolution imaging of NT-Lys and D4 in the absence or presence of Gag expression. To this end, we carried out two-color STORM imaging of HeLa cells labeled with JF549-D4 and AF647-NT-Lys 24 h after transfection with a vector or Gag/Gag-EGFP (Supplementary Fig. 15). We calculated the MOC values of the two fluorophores by Coloc-Tesseler implemented in the PoCA program to assess the extent of colocalization between JF549-D4 and AF647-NT-Lys. As a result, while 19% of AF647-NT-Lys colocalized with 35% of JF549-D4 in the absence of Gag, 36% of AF647-NT-Lys colocalized with 49% of JF549-D4 when Gag was expressed (Fig. 6g, h). This result strengthened our FLIM-FRET conclusion that Gag expression induces SM-rich and Chol-rich domains to be in close proximity.

## Membrane curvature affects the apposition of SM-rich and Chol-rich domains

Since the intrinsic curvature of the assembly sites at the PM continuously increases as Gag accumulates[36,38], the re-organization of SM/Chol domains may be due to a change in the membrane curvature at the assembly sites. To investigate the effect of PM curvature on the reorganization of lipid domains, we used the Gag-P99A and Gag-EE mutants that arrest the assembly process at a pre-budding or early budding step and form a relatively flat platform after their multimerization[40,41,81]. We also used the Gag-ΔL mutant that shows a defect in the recruitment of the endosomal sorting complex required for transport (ESCRT) machinery[40,41]. Gag-ΔL does not prevent changes in PM curvature but generates particles that remain attached to the PM. Therefore, unlike the wild-type Gag, which continuously releases particles, this mutant was expected to highlight a FRET population that corresponds to the curvature obtained at a late budding step.

The $\tau_1$ and $\alpha_1$ values in Gag-P99A and Gag-EE were similar to those in DA and WM mutants whereas those in Gag-ΔL were close to Gag WT (Fig. 6b, c). The FLIM diagram plots with Gag-P99A and Gag-EE showed a bimodal distribution similar to those without Gag and with Gag-WM

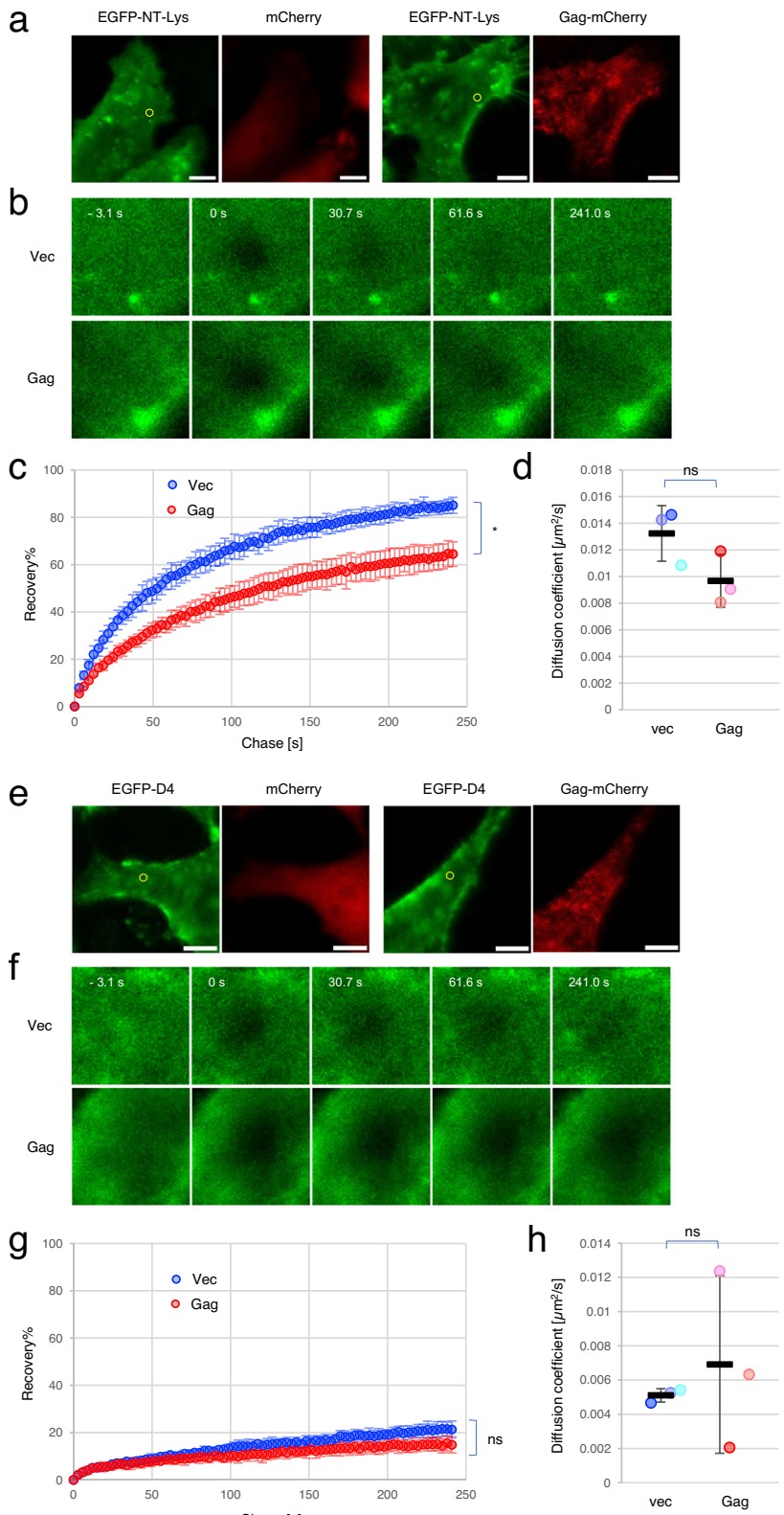

(Supplementary Fig. 13). In contrast, the FLIM plot of Gag-ΔL showed a pattern similar to that of Gag WT, which might reflect the accumulation of particles in late budding step with the Gag-ΔL mutant.

When looking at confocal images, Gag-ΔL-mTagBFP2 was unevenly localized in the PM and strongly correlated with the distribution of the shorter lifetimes like Gag WT-mTagBFP2, whereas Gag-P99A-mTagBFP2 and Gag-EE-mTagBFP2 mutants showed a relatively uniform localization in the PM and an independent localization from the lifetime distribution like Gag-WM-mTagBFP2 (Fig. 6d). We thus quantified the correlations between mTagBFP2 intensity and either $\tau_1$ or $\alpha_1$ in cells expressing these mutants as described above. We found negative or positive correlations between the intensity and $\tau_1$ or $\alpha_1$ in ΔL mutant-expressing cells and a slight negative correlation between the intensity and $\tau_1$ in EE mutant-expressing cells. Otherwise, there was

**Fig. 4 | Gag expression restricts the diffusion of EGFP-NT-Lys, and EGFP-D4 domains are more immobile than EGFP-NT-Lys domains. a, e** Representative images of cells expressing mCherry (vec) or Gag/Gag-mCherry (Gag) and labeled with EGFP-NT-Lys (**a**) and EGFP-D4 (**e**). Bar, 10 μm. **b, f** Fluorescent recoveries over time in the regions enclosed with yellow circles in (**a**) and (**e**). The number on the upper left side of each image indicates a chase time [s] from the time (t = 0) the fluorescence was bleached. **c, g** Averaged recovery curves of EGFP-NT-Lys (**c**) and EGFP-D4 (**g**) from three independent experiments are shown. Plots in blue and red indicate the values in vec and Gag, respectively. The *x*- and *y*-axes indicate chase time [s] and recovery %, respectively. The fluorescent recovery in each time point is shown as the mean ± SD in the three experiments. **d, h** Distributions of diffusion coefficients. The mean in each experiment and the mean ± SD in the three experiments are shown as dots in three different colors and black bars with error bars, respectively (*n* = 29 and 28 for vec and Gag in EGFP-NT-Lys, and 30 and 25 for vec and Gag in EGFP-D4 over three experiments). * or ns indicate a significant difference at the significance level of 0.05 or no significant difference, respectively, in the paired two-tailed *t*-test. Source data and *p*-values for (**c, d, g,** and **h**) were provided as a Source Data file.

no clear correlation in the other curvature-deficient mutants (Fig. 6e, f).

We further examined whether the curvature changes in mutant Gag assemblies impacted the sizes of SM-rich lipid domains by PALM/dSTORM imaging (Fig. 7a and Supplementary Fig. 16). We chose to examine SM-rich lipid domains by super-resolution microscopy because the mean diameter of Chol-rich lipid domains labeled with D4 did not show clear differences regardless of Gag expression (Fig. 3d). We analyzed the colocalization between Gag-mEos4b derivatives and AF647-NT-Lys and the domain sizes of AF647-NT-Lys and Gag-mEos4b derivatives as described earlier. In the colocalization analyses, ΔL demonstrated the same level (55%) of colocalization of Gag-mEos4b to AF647-NT-Lys as WT, which clearly differed from WM (25%) (Fig. 7b). The curvature-deficient P99A and EE mutants showed intermediate levels of colocalization (39% and 41%). The MOCs of AF647-NT-Lys to Gag-mEos4b derivatives showed small, non-significant differences between WT/ΔL and the other mutants (Fig. 7c). In addition, the mean (±SEM) diameter (220 ± 20 nm) of AF647-NT-Lys domains in Gag ΔL-expressing cells was similar to that in Gag WT-expressing cells (220 ± 10 nm) and significantly larger than those in vector-control (170 ± 10 nm) and Gag-WM (180 ± 20 nm) (Fig. 7d). AF647-NT-Lys in Gag-P99A and EE mutants showed mean domain diameters (200 ± 20 and 190 ± 10 nm) intermediate between vector-control (and WM) and ΔL (and WT). All these results indicate that Gag-ΔL mutant like Gag WT induces the formation of large SM-rich domains. Similar to AF647-NT-Lys domains, Gag-ΔL/Gag-ΔL-mEos4b and Gag WT/Gag WT-mEos4b formed domains with diameters around 180 nm. The domains formed by these two proteins were on average larger by 50–60 nm as compared to those formed by the other mutants (Fig. 7e). The conclusion on the differences in the sizes of the different protein domains is further confirmed through the distribution of the individual domains (Supplementary Fig. 10d and Supplementary Table 3).

The above results indicated that the ΔL mutant and Gag WT exhibit similar phenotypes in terms of lipid and protein domain formation. Some parameters, such as the amplitude $\alpha_1$ of the FRET population between EGFP-NT-Lys and mCherry-D4 were even higher for the ΔL mutant as compared to Gag WT. In contrast, the curvature-deficient P99A and EE mutants showed intermediate phenotypes between WT (and ΔL) and the multimerization-deficient WM mutant resembling the vector control. These findings suggest that the membrane curvature formed by Gag plays a key role in forming larger lipid domains.

## Discussion

It is not well understood how HIV-1 acquires specific lipids from the host PM. In this study, to address this question, we employed a non-toxic truncated mutant of lysenin, NT-Lys[30], and a cholesterol-binding domain of PFO, D4[45], to visualize endogenous cell surface SM- and Chol-rich lipid domains, respectively in Gag-expressing HeLa cells. Similar to the native lysenin[57,82], NT-Lys binds pre-existing SM clusters and does not induce clustering of SM[30,58]. Therefore, an absence of NT-Lys binding indicates a lack of SM clusters and not of SM[57]. Membrane binding of D4 requires very high Chol concentration (>40 mol%)[59,60,83]. This high Chol concentration threshold and the narrow dynamic range of these probes are useful to detect small changes in the membrane

Chol concentration[84–87]. Liposome binding experiments showed that EGFP-D4 equally bound brain SM/Chol and egg PC/Chol[59].

Our results indicate that (1) Gag expression does not alter the metabolism of SM and Chol; (2) Gag protein bound to the cytoplasmic leaflet of HeLa cell PM colocalized with endogenous SM-rich lipid domains or Chol-rich lipid domains of the outer leaflet at the similar proportion; (3) Gag restricted the lateral diffusion of SM-rich domains, while Chol-rich domains are intrinsically immobile compared to SM-rich domains; (4) Gag multimerization induced an increase in the size of SM-rich domains but not of Chol-rich domains; (5) Gag expression induced SM-rich and Chol-rich domains to be in close proximity in a curvature-dependent manner.

The Gag-induced reorganization of SM/Chol domains may be due to Gag-induced SM and/or Chol metabolism and subsequent changes in PM lipid composition or to Gag-induced changes in the lipid distribution of the PM. Figure 1 indicates that Gag expression did not significantly alter SM and Chol contents, and the total lipid composition of HeLa cells. Thus, Gag-induced reorganization of SM/Chol domains is unlikely due to changes in SM and Chol metabolism.

Our PALM/dSTORM results indicate that 26% of AF647-NT-Lys colocalized with 55% of Gag WT-mEos4b, while 27% of AF647-D4 colocalized with 52% of Gag WT-mEos4b. Different percentages of Gag and lipid probes may indicate the relative abundance of the number of lipid domains and Gag domains. It is a matter of debate whether Gag binds to pre-existing lipid domains or induces the formation of the lipid domains. Previously, we showed by PALM/dSTORM the transbilayer colocalization of SM-rich lipid domains in the outer leaflet and PI(4,5)P$_2$-rich lipid domains in the inner leaflet of the PM using NT-Lys and the PH domain of PLCδ that binds PI(4,5)P$_2$[46]. In addition, labeling of the PM with the SM/Chol binding protein, nakanori, showed transbilayer colocalization of SM/Chol domains and PI(4,5)P2-rich lipid domains labeled with PLCδ PH domain[32]. The transbilayer colocalization between Gag domains and SM- or Chol-rich domains observed in the current study can be explained at least in part by Gag binding to pre-existing PI(4,5)P$_2$- and SM- or SM/Chol-rich lipid domains. Moreover, our data also indicated that Gag multimerization induced lipid domains to form larger SM-rich lipid domains (Figs. 3, 5, and 7) and reorganization between SM-rich and Chol-rich domains (Fig. 6). Therefore, our study suggests that Gag proteins target pre-existing lipid domains and induce the formation of larger lipid domains.

FRAP experiments indicated that the expression of Gag significantly decreased the mobile fraction of SM-rich domains, and Chol-rich domains were intrinsically immobile even in the absence of Gag (Fig. 4). These observations are consistent with PALM/dSTORM experiments showing that the expression of Gag induced the formation of larger SM-rich domains but not the formation of larger Chol-rich domains. The formation of large lipid domains may be due to the coalescence of smaller lipid domains (Fig. 3, Supplementary Figs. 10, and 11). It is speculated that the association of Gag with SM-rich domains restricts the lateral diffusion of the lipid domains, resulting in the increase of the immobile fraction in FRAP measurements. The formation of larger lipid domains could be driven by trapping the mobile fraction of SM-rich domains to the domains where Gag is assembling.

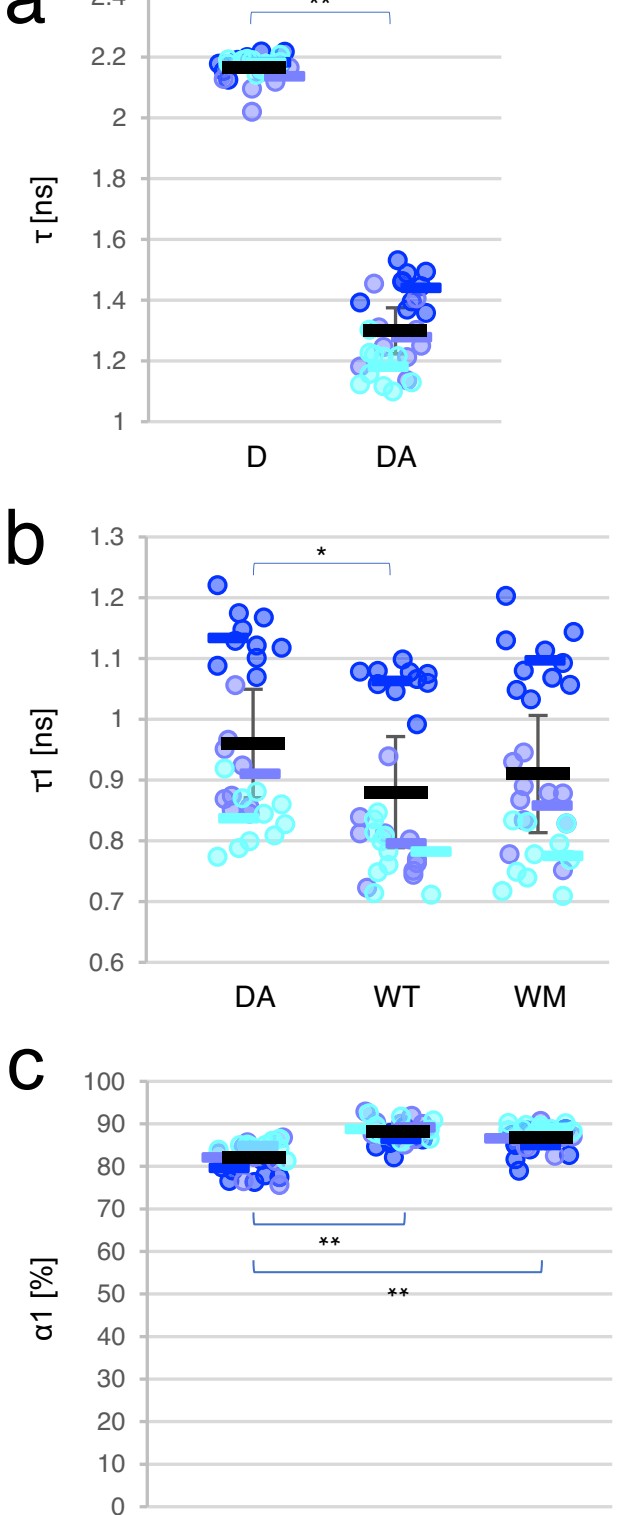

**Fig. 5 | Expression of Gag WT and WM mutant caused a reorganization of SM-rich domains in the PM. a** Lifetime distribution in the cell labeled with EGFP-NT-Lys (donor, D) and EGFP-NT-Lys and mCherry-NT-Lys (acceptor) (DA) in FLIM-FRET. The results of the single-component analysis were shown. **b, c** Lifetime $\tau_1$ (**b**) and amplitude $\alpha_1$ (**c**) distribution in the two-component analysis of cells doubly labeled with EGFP-NT-Lys and mCherry-NT-Lys. DA, WT, and WM indicate cells without and with Gag WT and Gag-WM expression, respectively. The value in each cell, the mean in each experiment, and the mean ± SEM in the three independent experiments are shown as dots and bars in three different blue colors, and a black bar and error bar, respectively ($n = 30$ except for DA in which $n = 29$ over three experiments). The two-tailed paired $t$-test is applied on (**a**) and the one-way ANOVA post-hoc Tukey test on (**b**) and (**c**). * and ** indicate significant differences at the significance level of 0.05 and 0.01, respectively. Source data and detailed statistics were provided as a Source Data file.

The wild-type Gag lattice is composed of hexamers interconnected via the dimer interface at the level of the CA domain in the immature viral particle[68]. The WM mutant carrying two mutations, W184A and M185A, in the dimer interface of CA domain[66–68,70], was used as a non-multimerizable mutant[9,70] in this study.

PALM/dSTORM results showed that Gag WT but not WM mutant induced the formation of larger SM-rich domains labeled with NT-Lys. FLIM-FRET experiments showed that both Gag WT and Gag-WM increased the population with high FRET efficiency between NT-Lys molecules. This difference in the WM-mutant results between PALM/dSTORM and FLIM-FRET may result from the difference in the lipid domains detected by these two methods. While FLIM-FRET analysis detected FRET occurred by close apposition of at least two NT-Lys molecules consisting of a donor and an acceptor, our PALM/dSTORM analysis considered a domain including 10 or more localizations of NT-Lys as a domain. Therefore, we speculate that the binding of WM mutant to the inner leaflet of the PM may induce the formation of small SM-rich domains and/or stabilize these domains due to the lack of multimerization.

We also measured FRET between SM-rich and Chol-rich domains. When HeLa cells were labeled with EGFP-NT-Lys and mCherry-D4 in the absence of Gag, FLIM-FRET plot showed two main interacting SM/Chol populations corresponding to peaks at $\tau_1 = 1.9$ ns and 1.1 ns (DA in Supplementary Fig. 13). The amplitude associated with the short-lived lifetime, 1.1 ns, suggests that there are about 20% of the SM clusters that are closely associated with D4-labeled Chol-rich domains (DA in Supplementary Fig. 13) and the rest 80% (not shown on the plot) that are not in FRET distances with D4-labeled Chol-rich domains in the outer leaflet of the PM. As the size of the confocal area (0.2 μm²), in which each decay curve is recorded, is orders of magnitude larger than the area occupied by a complex of a few SM/Chol molecules (>10 nm²), it can be speculated that each confocal area contains a large number of SM-rich and Chol-rich domains. The above data suggests the presence of heterogeneous complexes or domains that may differ by the arrangement and/or packing of SM and Chol molecules. The existence of SM/Chol complexes at the cell surface of HeLa cells is fully in line with the previously observed labeling of HeLa cells by nakanori, a lipid binding protein that specifically binds to SM/Chol complexes[32].

When Gag WT was expressed in HeLa cells, the FRET population peak shifted from 1.9 to 1.2 ns in $\tau_1$ (WT in Supplementary Fig. 13), suggesting that Gag bound to the cytoplasmic leaflet of the PM reorganizes the SM/Chol-rich lipid domains in the outer leaflet. The amplitude associated with this short-lived lifetime suggests that the percentage of SM clusters involved in domains increased to ~25%. This is consistent with the increase in the domain diameter (Fig. 3) and the colocalization of SM-rich domains and Chol-rich domains (Fig. 6). In addition, Gag WT expression reduced the mean density of AF647-NT-Lys and AF647-D4 in SM-rich and Chol-rich domains, respectively (Fig. 3). This reduction in the probe density might be explained by an increase in the FRET population where EGFP-NT-Lys and mCherry-D4 got in close

The diffusion coefficient that we obtained for the SM-rich domains (0.013 ± 0.002 μm²/s) is 10-fold slower than that obtained by a fluorescent SM analog, ATTO647N-SM, and STED-FCS[42]. This discrepancy may be explained by the difference in these probes. ATTO647N-SM is a lipid analog that has a charged bulky fluorophore attached to the acyl chain[44,88], whereas NT-Lys recognizes a SM cluster consisting of 5–6 molecules of SM[57,89].

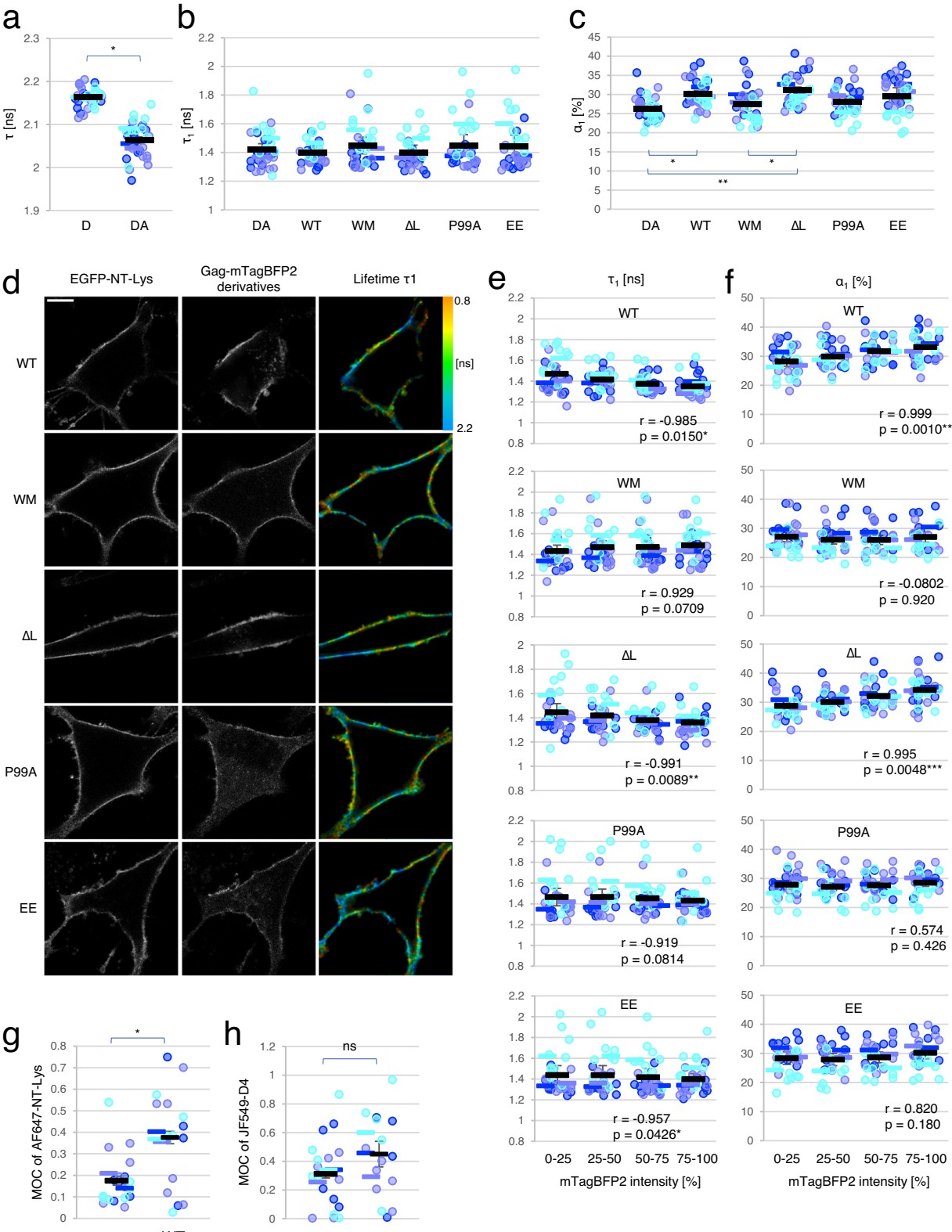

proximity. The Gag-WM mutant did not impact the reorganization of SM-rich and Chol-rich domains, suggesting that Gag multimerization is crucial for promoting the interaction of SM-rich and Chol-rich domains.

In this study, we further examined the effects of the membrane curvature induced by Gag mutants on the redistribution of SM-rich and Chol-rich domains measured by FLIM-FRET as well as on the colocalization and the sizes of SM-rich and Gag domains measured by PALM/

dSTORM. P99A and EE (E75A E76A) mutants form a relatively flat platform upon their multimerization[40,41,90] due to the weak interaction between Gag proteins or Gag and host factors[67,91]. The inefficient coalescence of SM-rich and Chol-rich domains in the presence of these mutants (Figs. 6 and 7, Supplementary Tables 1–3) underlines the importance of membrane curvature for the reorganization of lipid domains. Our data are in variance with the observation that the

**Fig. 6 | Expression of Gag caused a reorganization of SM-rich and Chol-rich domains in the PM. a** Lifetime distribution in the cells labeled with EGFP-NT-Lys (D), EGFP-NT-Lys and mCherry-D4 (DA) in FLIM-FRET. The lifetime τ [ns] in a single-component analysis was shown. **b, c** Lifetime $τ_1$ [ns] (**b**) and amplitude $α_1$ [%] (**c**) of cells doubly labeled with EGFP-NT-Lys and mCherry-D4. **b**–**f** DA, WT, WM, ΔL, P99A, and EE indicate cells expressing vectors, Gag WT/Gag WT-mTagBFP2, Gag-WM/Gag WM-mTagBFP2, Gag-ΔL/Gag-ΔL-mTagBFP2, Gag-P99A/Gag-P99A-mTagBFP2, and Gag-EE/Gag-EE-mTagBFP2, respectively. **d** Representative images of EGFP-NT-Lys, Gag-mTagBFP2 derivatives, and lifetime $τ_1$ in WT, WM, ΔL, P99A, and EE. $τ_1$ [ns] from 0.8 to 2.2 is indicated in pseudo-colors on the right. Bar, 10 μm. **e, f** Mean $τ_1$ [ns] (**e**) or mean $α_1$ [%] (**f**) in WT, WM, ΔL, P99A, and EE were plotted in every quartile of mTagBFP2 intensity. Pearson correlation coefficients were calculated between mean $τ_1$ or $α_1$ in each range and the rank of the intensity (see "Methods"). The r and

p (two-tailed) values were indicated in the graphs. *, **, and *** indicate that there is a correlation at the significance level of 0.05, 0.01, and 0.005. **g, h** MOC values between AF647-NT-Lys and JF549-D4 in dSTORM imaging of vector (vec) or Gag WT/Gag WT-mEos4b (WT) -transfected cells. **g** MOCs of AF647-NT-Lys to JF549-D4, **h** MOCs of JF549-D4 to AF647-NT-Lys. The value in each cell, the mean in each experiment, and the mean ± SEM in the three independent experiments are shown as dots and bars in three different blue colors, and a black bar and error bar, respectively ($n = 30$ for all except for EE ($n = 29$) (**a**–**f**), and $n = 18$ and 15 for vec and WT (**g** and **h**), over three experiments). The two-tailed paired $t$-test is applied on (**a**), (**g**), and (**h**), and the one-way ANOVA post-hoc Tukey test on (**b**) and (**c**). * and ** indicate significant differences at the significance levels of 0.05 and 0.01, respectively. ns indicates not significant. Source data and detailed statistics were provided as a Source Data file.

enrichment of exogenously added fluorescent SM analog, N-[6-((7-nitrobenz-2oxa-1,3-diazol-4-yl)amino)hexanoyl]-sphingosine-1-phosphocholine (NBD-SM), in the assembly sites, was not affected by P99A mutant expression[41]. This may be due to the difference in physical properties between the fluorescent lipid analog and natural lipids[43,44,92]. NBD-SM has been shown to mainly distribute in Ld (liquid disordered) domains in model membranes due to the hydrophilic NBD[43] while in contrast, NT-Lys or lysenin-positive SM clusters are distributed in Lo (liquid ordered) domains[58]. In contrast to P99A and EE, the curvature-forming Gag-ΔL mutant showed similar results to Gag WT. These results suggest the importance of membrane curvature for the reorganization of endogenous lipid domains.

In conclusion, our results indicate that HIV-1 Gag expressed in the cytoplasm altered the SM-rich and marginally Chol-rich domains, likely by bringing pre-existing domains with different lipid compositions close together and further reorganizing these domains to form larger domains. This conclusion is in line with the data of Ono's group reporting that Gag induced coalescence of lipid raft domains and tetraspanin-enriched domains[40]. Recently, Sengupta et al. reported that proteins were either recruited into or removed from the HIV-1 assembly sites through lipid-based partitioning, initiated by Gag multimerization and amplified by changes in membrane curvature at the assembly site[41]. Merging of lipid raft proteins and tetraspanins was also reported to be less efficient when a curvature-deficient Gag was expressed[40]. In this study, curvature-deficient Gag-P99A and -EE mutants exhibited inefficient domain merging (Figs. 6 and 7). These results suggest that lipid reorganization by Gag is also curvature-dependent. Moreover, an additional lipid reorganization induced by Gag occurred within coalesced domains as suggested by the formation of larger but less dense lipid domains (Figs. 3 and 6) and the increased immobile fraction of SM-rich domains (Fig. 4). Our results and previously published results[40,41] suggest that Gag-oligomerization- and curvature-dependent reorganization of lipid domains may cause the partitioning of proteins favoring specific lipid environments.

## Methods

### Cells

The HeLa cell line was obtained from the American Type Culture Collection (CCL-2, ATCC, VA). Cells were maintained in Dulbecco's modified Eagle's medium (DMEM; 2188-025, Gibco) supplemented with 10% fetal bovine serum (FBS, Lonza, Switzerland), 100 U/mL penicillin, and 100 μg/mL streptomycin (DMEM/FBS/P/S) at 37 °C. Plasmid constructs were transfected into HeLa cells using jetPRIME reagent (Polyplus transfection, France). For cell labeling with fluorescent lipid probes, DMEM supplemented with 5% lipoprotein-deficient serum (DMEM/LPDS; S5394, Sigma-Aldrich) was used.

### Construction, expression, and purification of lipid probes

For the construction of six histidine-tagged (His6-)SNAP-NT-Lys and His6-SNAP-D4 plasmids, EGFP and mCherry sequence in pET28/EGFP-NT-Lys and pET28/mCherry-D4[46] were replaced with SNAP sequence

amplified by PCR using pSNAPf vector (New England Biolabs, MA) as a template and primers (5'-GCGGTACCATGGACAAAGACTGCGAAATG-3' and 5'- TTAAGCTTACCCAGCCCAGGCTTGCC-3'). Expression and purification of lipid probes were performed according to Tomishige et al.[93]. Briefly, expressions of His6-tagged NT-Lys and D4 were induced in *E.coli* BL21(DE3) by culturing at 18 °C for two overnights and 18 h in the presence of 250 μM and 125 μM isopropyl β-D-1-thiogalactopyranoside, respectively. Lipid probes were bound to a HiTrap TALON crude column (29-0485-65, GE Healthcare) after lysing bacteria in BugBuster Master Mix (71456, Novagen) supplemented with protease inhibitor cocktail set I (539131-10VL, Calbiochem), because of the presence of His6 tag at their N-termini. Bound lipid probes were eluted with phosphate buffer containing 400 mM imidazole and dialyzed in PBS. Purified lipid probes were stored as 50% glycerol solution at −20 °C after the measurement of protein concentration. Labeling of His6-SNAP-NT-Lys and His6-SNAP-D4 with Alexa Fluor 647 was performed using SNAP-Surface Alexa Fluor 647 (S9136S, New England Biolabs) by following the manufacturer's protocol before each experiment. Labeling of His6-SNAP-D4 with Janelia Fluor 549 was performed using JF549cp-SNAP-tag ligand (Janelia Materials, VA) in the same manner as the labeling with AF647.

### Preparation of Gag and Gag mutant plasmids

The expression plasmid of pcDNA/Gag-mCherry was prepared as described[16]. In all the fluorescent protein (FP) tagged Gag plasmids derived from this plasmid, the FP sequence is inserted at the C-terminus of the MA domain as described below. The plasmid pcDNA/Gag for expression of non-tagged wild-type Gag was constructed by deleting the mCherry sequence and restriction enzyme sites at both ends from pcDNA/Gag-mCherry by PCR using the primer pairs (5'-GTGAGCCAGAACTACCCCATCGTGCAGAAC-3' and 5'-CTGGCTGCTGTTGCCGGTGCCGGC-3'). The plasmids pcDNA/Gag-mTagBFP2 and pcDNA/Gag-WM-mTagBFP2 were constructed by replacing mCherry in pcDNA/Gag-mCherry and pcDNA/Gag-WM-mCherry with mTagBFP2 amplified by PCR using mTagBFP2-Lifeact-7 as a template and primers (5'-TA<u>GGATCC</u>ATGGTGTCTAAGGGCGAAGAGCTG-3' and 5'-CC<u>GAATTC</u>ATTAAGCTTGTGCCCCAGTTTGCTAGG-3'). The plasmid pcDNA/mTagBFP2 was constructed by cloning the mTagBFP2 sequence amplified in the above PCR into respective sites in the pcDNA3 vector. The plasmids, pcDNA/Gag-P99A, pcDNA/Gag-EE, and pcDNA/Gag-ΔL, were constructed by PCR mutagenesis using pcDNA/Gag as a template and primers (5'-GCCCGCGGCAGCGACATCGCCGGC-3' and 5'-CTCGCGCATCTGGCCGGGGGCGATGG-3' for Gag-P99A; 5'- GCCGCCGAGTGGGACCGC −3' and 5'- CGCCGCGTTGATGGTCTCCTTCAGCATCTGCATGG −3' for Gag-EE; 5'-GCCGCCGCCGCGCCCCCGAGGAGAGCTTCCGCTTCGGC-3' and CTCGGGGCGGCTCTGCAGGAAGTTGCCGGGGCGGCC-3' for Gag-ΔL). The plasmids, pcDNA/Gag-P99A-mTagBFP2, pcDNA/Gag-EE-mTagBFP2, and pcDNA/Gag-ΔL-mTagBFP2, were constructed in the same manner as the non-tagged counterparts except that pcDNA/Gag-mTagBFP2 was used as a template. The plasmids encoding mEos4b-fusion of Gag derivatives were constructed by

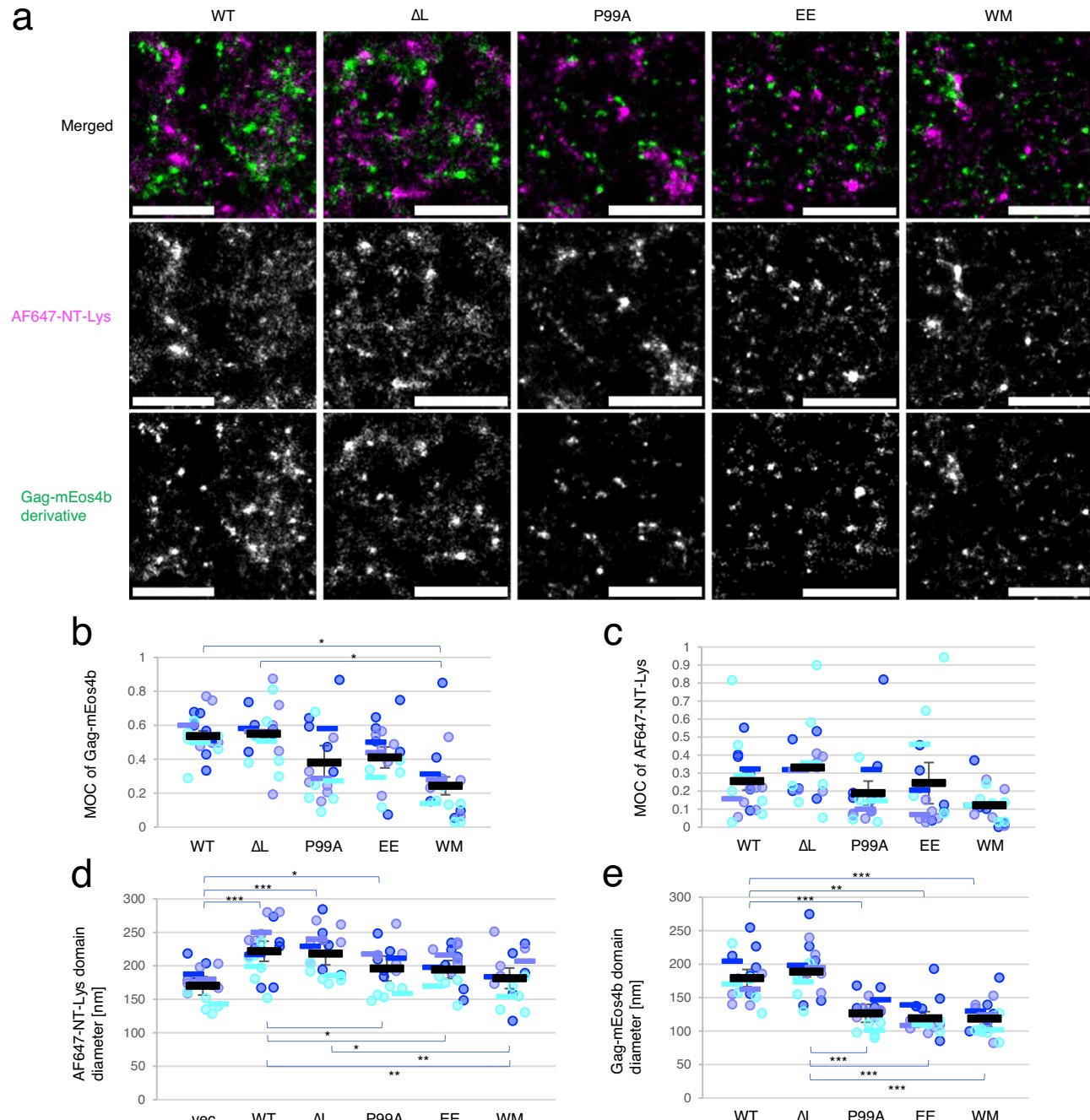

**Fig. 7 | Colocalization between Gag-mEos4b curvature mutants and AF647-NT-Lys and domain analysis in PALM/dSTORM. a** Representative images of Gag/Gag-mEos4b derivatives-expressing HeLa cells labeled with AF647-NT-Lys. From top to bottom: merged, AF647-NT-Lys, and Gag-mEos4b derivatives images. Bar, 1 μm. Vec, cells expressing an empty vector; WT, Gag WT/Gag WT-mEos4b; ΔL, Gag-ΔL/Gag-ΔL-mEos4b; P99A, Gag-P99A/Gag-P99A-mEos4b; EE, Gag-EE/Gag-EE-mEos4b; WM, Gag-WM/Gag-WM-mEos4b. **b** MOC values in colocalization analysis of Gag-mEos4b derivatives to AF647-NT-Lys. **c** MOC values of AF647-NT-Lys to Gag-mEos4b derivatives. **d, e** the mean diameters [nm] of AF647-NT-Lys (**d**) and

Gag-mEos4b derivatives (**e**) domains in each cell, the means in each of three independent experiments, and the mean ± SEM were shown as dots and bars in three different blue colors, and black bars with the error bars ($n = 18, 17, 16, 16$, and 16 for WT, ΔL, P99A, EE, and WM over three experiments). *, **, and *** indicate significant differences at the significance levels of 0.05, 0.01, and 0.005 in the one-way ANOVA post-hoc Tukey test, respectively. For comparison, the data of vec, WT, and WM samples in Figs. 2c–f, 3a, c were shown again. Source data and detailed statistics for (**b**–**e**) were provided as a Source Data file.

replacing mTagBFP2 with mEos4b sequence amplified by PCR using mEos4b-C1 as a template and primers (5′-ACGGATCCATGGT-GAGTGCGATTAAGCCAGAC-3′ and 5′-TTGAATTCTCGTCTGG-CATTGTCAGGCAATC-3′). The absence of unwanted mutations was confirmed by DNA sequencing (Eurofins Genomics, Germany). Oligonucleotide sequences used in this study were synthesized in and

purchased from Eurofins Genomics. The plasmid mTagBFP2-Lifeact-7 (Addgene plasmid #54602; http://n2t.net/addgene: 54602; RRID: Addgene_54602) and mEos4b-C1 (Addgene plasmid #54812; http://n2t.net/addgene: 54812; RRID: Addgene_54812) were gifts from Michael Davidson. The plasmids constructed in this study are readily available from the authors.

## Thin-layer chromatography analysis of sphingolipids, phospholipids and neutral lipids

In the thin-layer chromatography (TLC) analysis, lipids were extracted by following Bligh and Dyer[94] from HeLa cell lysate equivalent to 100 μg of protein after 24 h of transfection with a 3:1 mixture of pcDNA and pcDNA/mTagBFP2 or non-tagged and mTagBFP2-tagged Gag plasmids. The extracted lipids were separated on high-performance thin layer chromatography (HPTLC; 1.05633.0001, Merck, Germany) plates in acetone/methanol/acetic acid/chloroform/water (30:26:24:80:10 by vol.) for polar lipids and hexane/diethylether/acetic acid (80:20:2 by vol.) for neutral lipids. After SM and Chol bands were visualized by spraying cupric acetate solution (3% copper acetate, 8% phosphoric acid, water by vol.) and heating at 180 °C for 5 min and by spraying ferric chloride reagent and heating at 120 °C for 3 min[95], respectively. The bands were imaged by LAS4000mini under the control of Image Reader LAS-4000 ver. 1.0 (GE Healthcare) and quantified by their densities using Multi Gauge ver. 3.1 (GE Healthcare). The position of each lipid on HPTLC was determined using lipid standards. Uncropped images of HPTLC plates are shown in the Source Data file.

## Confocal microscopy

HeLa cells were seeded in 0.5 mL of DMEM/FBS/P/S at a density of $0.6 \times 10^5$ cells/mL in a well of a 24-well plate with a 12-mm glass coverslip one day before transfection. The next day, the cells in the well were transfected with 0.25 μg of a 3:1 mixture of non-tagged and mTagBFP2-tagged Gag plasmids using jetPRIME reagent according to the manufacturer's protocol. After 24 h of the transfection, the transfected cells were sequentially labeled with 1.2 μM mCherry-D4 and then 0.2 μM EGFP-NT-Lys at 37 °C for 15 min each by washing twice with DMEM/LPDS between the two labelings. After fixing the labeled cells with 4% paraformaldehyde (PFA; 15710, Electron Microscopy Sciences) containing 10 μM Hoechst33342 (62449, Thermo Scientific, MA) at r.t. for 30 min and neutralizing the residual PFA, the coverslips were mounted using a drop of ProLong Diamond (P36965, Thermo Scientific) on a glass slide. The slides were kept in the dark until the mounting medium hardened. The slide was observed under the microscope LSM700 (Zeiss, Germany) equipped with a C-Apochromat 63×W Corr (1.2 NA) objective (Zeiss). Fluorescent images were acquired using 405, 488, and 555 nm lasers for Hoechst33342, EGFP, and mCherry excitation through manipulation on ZEN2008 (Zeiss) software. Representative images are shown in Fig. 2a.

## Fluorescence recovery after photobleaching (FRAP)

HeLa cells were seeded in 2 mL of DMEM/FBS/P/S at a density of $0.5 \times 10^5$ cells/mL in a sealable 35-mm glass bottom dish (μ-Dish; 81158, μ-Dish$^{35 mm, high}$ Glass Bottom: Φ 35 mm, high wall, # 1.5H, ibidi GmbH, Germany). The next day, the cells in the dish were transfected with 1 μg of a 3:1 mixture of pcDNA/Gag WT and pcDNA/Gag-mCherry, or pcDNA3.1 and pmCherry-N using jetPRIME reagent 22 h before the experiment according to the manufacturer's protocol. The cells were labeled with 0.2 μM EGFP-NT-Lys or 1.2 μM EGFP-D4 in DMEM/LPDS at 37 °C for 15 min. After replacing the labeling medium with DMEM/LPDS, the cells expressing Gag-mCherry or mCherry alone were selected under the confocal microscope LSM700 equipped with a C-Apochromat 63×W Corr (1.2 NA) objective. Around every 3 s, 10 and 79 frames were recorded before and after photobleaching with a 488 nm laser at a spot of 2 μm diameter. All these settings and manipulations of the microscope were performed through ZEN2008 (Zeiss) software. The cell labeling and acquisition took approximately 1.5 h in each condition. The image processing and analysis were performed as described in refs. 96,97. In brief, after processing raw data in each measurement, the averaged fluorescence recovery in each experiment was fitted using the program, easyFRAP[97] (stand-alone ver., http://ccl.med.upatras.gr/tools-easyfrap/) and MATLAB (R2023a,

Mathworks, MA) to run the program, and the diffusion coefficient was calculated after calculating half-recovery time in easyFRAP. To draw a recovery curve, the mean and standard deviation at each time point were calculated from three means of the three independent experiments ($n = 7$–11 in each) and plotted as a function of chase time.

## PALM/dSTORM

HeLa cells were seeded in 2 mL of DMEM/FBS/P/S at a density of $0.5 \times 10^5$ cells/mL in a sealable 35-mm μ-Dish one day before transfection. The next day, the cells in the dish were transfected with 1 μg of a 3:1 mixture of non-tagged and mEos4b-tagged Gag plasmids using jetPRIME reagent according to the manufacturer's protocol. The transfected cells were labeled with 0.2 μM AF647-NT-Lys or 0.8 μM AF647-D4 in DMEM/LPDS at 37 °C for 15 min after 24 h of transfection. For labeling with two lipid probes at 37 °C for 15 min each, cells were first incubated with 0.4 μM JF549-D4 in DMEM/LPDS, then followed by the labeling with 0.2 μM AF647-NT-Lys. The labeled cells were washed with PBS twice and fixed with 4% paraformaldehyde (PFA) at room temperature for 30 min. After neutralizing the residual PFA, the fixed cells on a dish were stored in PBS at 4 °C by the time of acquisition. PALM/dSTORM acquisition was performed on the cells in the STORM buffer (10% (w/v) glucose, 0.56 mg/mL glucose oxidase, 0.17 mg/mL catalase, and 100 mM β-mercaptoethylamine) added just before the acquisition by using a home-built set up based on a Nikon Eclipse Ti microscope with 100× 1.49 NA oil-immersion objective, ImageEM X2 EM-CCD camera (Hamamatsu Photonics, Japan), and W-VIEW GEMINI image splitting optics (Hamamatsu Photonics) as described[98]. Image acquisition was sequentially carried out in the red (AF647) and the green (mEos4b or JF549-D4) channel, either 12,000 frames each for AF647 (13.9 ms/frame) and mEos4b (30.5 ms/frame) combination or 20,000 frames each for AF647 and JF549 combination (13.9 ms/frame for both). Fluorescence emission was imaged by using a 642 nm laser for AF647, 561 nm laser for JF549, and both 405 nm and 561 nm lasers for mEos4b photoconversion and excitation, respectively. Each time before sample acquisition, we recorded a raster scan of a single TetraSpek bead (Molecular Probes, OR) in two channels to later correct for chromatic aberration. The microscope system was controlled by μManager ver. 1.4.22[99]. Acquired image data were analyzed by Detection of Molecules (DoM) plugin v.1.1.6 (https://github.com/UU-cellbiology/DoM_Utrecht) for Fiji (ver. 2.14.0/1.54f) to detect localization coordinates and correct them for chromatic aberration. The obtained coordinates were imported to ThunderSTORM (ver. 1.3)[100] plugin for Fiji to further filter them (120 nm <σ (standard deviation of Gaussian fit) <180 nm, intensity > 100, and chi2 > 0.5 (for AF647 and mEos4b) or chi2 > 0.2 (for JF549)), correct for drifting using the cross-correlation method, and export it as a new coordinate file for the domain analysis. The new coordinates were imported into and analyzed by PoCA ver. 0.5.0 (Point Cloud Analysis; https://github.com/flevet/PoCA)[73] that integrates most of the functions of SR-Tesseler[75] and Coloc-Tesseler[72] to detect AF647-NT-Lys domains, AF647-D4 domains, JF549-D4 domains, and Gag-mEos4b domains, and calculate sizes of these domains and the number of these fluorescent lipid probes and Gag-mEos4b localizations included in the domains, and MOC as the indication of colocalization between two fluorophores. The data were presented according to ref. 101. The localization uncertainty (Supplementary Note) was calculated from the data exported from ThunderSTORM.

## FLIM-FRET

**Sample preparation.** HeLa cells were seeded in 2 mL of DMEM/FBS/P/S at a density of $0.5 \times 10^5$ cells/mL on a sealable 35-mm μ-Dish one day before transfection. The next day, the cells in the dish were transfected with 1 μg of a 3:1 mixture of non-tagged and mTagBFP2-tagged Gag plasmids using jetPRIME reagent according to the manufacturer's

protocol. After 24 h of transfection, the transfected cells were labeled with 0.2 μM EGFP-NT-Lys in DMEM/LPDS at 37 °C for 15 min. In the case of double labeling, the cells were sequentially labeled with 1.2 μM mCherry-D4 and then 0.2 μM EGFP-NT-Lys at 37 °C for 15 min each by washing twice with DMEM/LPDS between the two labelings. After fixing the labeled cells with 4% PFA at r.t. for 30 min and neutralizing the residual PFA, the cells in the dish were stored in PBS at 4 °C by the time of acquisition.

**Acquisition.** FLIM measurements were performed at the middle plane of a cell prepared above in PBS with a homemade two-photon excitation scanning microscope based on an inverted microscope (IX70, Olympus) with a 60× 1.2 NA water immersion objective operating in the descanned fluorescence collection mode[102]. The EGFP was excited at 930 nm using a femtosecond laser (Insight DeepSee, Spectra Physics). Fluorescence photons were collected at a dwell time of 4 μs/pixel using a short-pass filter with a cutoff wavelength of 680 nm (F75-680, AHF) and a band-pass filter of 525/40 nm (F37-520, AHF). The fluorescence was directed to a fiber-coupled APD (SPCM-AQR-14-FC, Perkin Elmer), which was connected to a time-correlated single photon counting module (SPC830, Becker & Hickl). The measurement was controlled by SPCM ver 9.83 (Becker & Hickl).

**Data analysis.** FLIM images were analyzed using a commercial software package (SPCImage V8.1, Becker & Hickl, Germany). A binning of two (5 × 5 pixels (pixel size was around 230 nm)) was applied before processing the fluorescence decays. The FLIM data were further analyzed to obtain the lists of $\tau_1$ and $\alpha_1$ in each pixel within a cell and the FLIM diagram plots, using a homemade R script named flimDiagRam (V.0.1: https://doi.org/10.5281/zenodo.8408457)[79,103,104]. In the two-component analysis, two populations are assumed to contribute to the EGFP-NT-Lys decay profile with one population consisting of non-transferring EGFP-NT-Lys molecules (more than 10 nm apart from an acceptor) and one population of EGFP-NT-Lys molecules with one or several acceptor molecules in close proximity (<10 nm), so that FRET can occur. Based on this assumption, the fluorescence decays can be fitted to a double exponential Eq. (1):

$$I(t) = I_0 \left( \alpha_1 e^{-\frac{t}{\tau_1}} + \alpha_2 e^{-\frac{t}{\tau_2}} \right) \tag{1}$$

where $\tau_1$ is the short-lived lifetime of the EGFP-NT-Lys population undergoing FRET and $\tau_2$ is the lifetime for the unquenched donors. The relative contribution of each population is given by $\alpha_1$ and $\alpha_2$, linked by $\alpha_1 = 1 - \alpha_2$. To extract $\tau_2$, the value of the donor only fluorescence lifetime, images of the cells labeled with EGFP-NT-Lys alone were analyzed with a single exponential equation. The means of the lifetimes of EGFP-NT-Lys alone were $2.17 \pm 0.20 \times 10^{-1}$ ns and $2.16 \pm 0.40 \times 10^{-2}$ ns for Figs. 5 and 6, respectively. By rounding these values and fixing $\tau_2$ at 2.2 ns, a pair of the lifetime and its amplitude ($\tau_1$, $\alpha_1$) in each pixel was obtained. These $\tau_1$ and $\alpha_1$ values were plotted to draw 1D distribution in Figs. 5 and 6 according to ref. 101. These pairs of $\tau_1$ and $\alpha_1$ were also plotted in 2D (FLIM diagram plot) to reveal the main tendencies as well as the distribution of the individual parameters. The above assumption was validated on the actual data as described in the Supplementary Note. The contour line in a FLIM diagram was determined by the kernel density estimate function in the flimDiagRam (Supplementary Note) indicates probability distribution. The FRET efficiency (E) in the text was calculated according to:

$$E = 1 - \frac{\tau_{DA}}{\tau_D} \tag{2}$$

where $\tau_{DA}$ and $\tau_D$ are the lifetimes of the donor in the presence and absence of the acceptor, respectively. To find a correlation between Gag-mTagBFP2 intensity and either lifetime $\tau_1$ and amplitude $\alpha_1$, we utilized a homemade imageJ-MATLAB program to extract $\tau_1$ and $\alpha_1$ values in an ROI corresponding to EGFP-NT-Lys signal from a dataset analyzed by SPCImage. The obtained values were sorted into quartiles of mTagBFP2 intensity and then plotted in every quartile. Pearson correlation coefficient was calculated between 4 means of the $\tau_1$ or $\alpha_1$ values and ranks of mTagBFP2 intensity using GraphPad Prism 5.04 (GraphPad Software, MA). In this analysis, we set the ranks (1 to 4) corresponding to 4 quartiles (0–25%, 25–50%, 50–75%, and 75%–100%) in order. Supplementary Fig. 17 shows that there were no correlations between the mean fluorescence intensity of Gag-mTagBFP2 and mean lifetime $\tau_1$ or its amplitude $\alpha_1$ in FLIM-FRET of EGFP-NT-Lys and mCherry-D4. Supplementary Fig. 17 suggests that the variation in the expression level of Gag derivatives in our experimental conditions did not significantly affect the Gag-induced lipid domain reorganization.

## Checking binding specificity of lipid probes newly developed in this study

HeLa cells were seeded in 0.5 mL of DMEM/FBS/P/S at a density of 0.3 × 10⁵ cells/mL in a well of a 24-well plate with a 12-mm glass coverslip one day before transfection. Next day, the cells in the well were labeled with 0.2 μM AF647-NT-Lys, 0.8 μM AF647-D4, or 0.4 μM JF549-D4 after treatment with mock and either 0.125 U bacterial SMase (bSMase; S8633, Sigma-Aldrich) or 10 mM methyl-β-cyclodextrin (MbCD; C4555, Sigma-Aldrich) at 37 °C for 30 min, for NT-Lys or D4 labeling, respectively, according to Tomishige et al.[93]. After fixing the labeled cells with 4% PFA containing 10 μM Hoechst33342 at r.t. for 30 min and neutralizing the residual PFA, the coverslip was mounted using a drop of ProLong Diamond on a glass slide. The slides were kept in the dark until the mounting medium hardened. The slide was observed under the microscope FV1000 (Olympus, Japan) equipped with a PlanApo 60×WLSM (1.0 NA) objective (Olympus). Fluorescent images were acquired using 405, 543 and 633 nm laser for excitation of Hoechst33342, JF549, and AF647, respectively under the control by FV10-ASW4.2 (Olympus) software. The representative images are shown in Supplementary Fig. 1.

## Western of VLP fraction and cell lysate

293T cells were seeded in DMEM/FBS/P/S at a density of 7.5 × 10⁵ cells/mL in a T75 flask one day before transfection and cultured at 37 °C for 20 h. The cells were transfected with a total of 5 μg of pcDNA3/Gag alone or a mixture of pcDNA3/Gag and pcDNA3/Gag-FP (3:1) using jetPRIME transfection reagent (Polyplus transfection, France) according to the manufacturer's instruction. After 24 h of the transfection, the culture medium was collected and centrifuged at 900×g for 5 min. The supernatant (sup) of the centrifuged medium was filtrated through a 45 μm filter unit (Millex-HP, SLHPM33RS, Millipore, Germany) and then the filtrated culture sup (around 10 mL) was transferred into an ultracentrifugation tube (25 × 89 mm, BECKMAN COULTER, CA). Five mL of TNE buffer (10 mM Tris-Cl, pH 7.4, 100 mM NaCl, and 1 mM EDTA) was added and 3 mL of 20% sucrose in TNE buffer was placed at the bottom of the tube. The samples set in the rotor 70 Ti (BECKMAN COULTER) were ultracentrifuged at 208,656×g (45,000 rpm) for 2 h at 4 °C in the ultracentrifuge (Optima LE-80K, BECKMAN COULTER). The pellet was dissolved and suspended in 100 μL of TNE buffer containing protease inhibitor cocktail (PIC; Protease inhibitor cocktail set I, 539131-10VL, Calbiochem). After collecting the culture medium, the cells attached to the surface of T75 were washed with PBS twice, collected with a scraper, and transferred into a 15 mL tube. The cells were suspended in 200 μL of SHEI buffer (10 mM HEPES-NaOH, pH 7.4, 250 mM sucrose, 1 mM EDTA, and PIC) and sonicated briefly. After mixing samples with 4× sample buffer and heating them at 95 °C for 5 min, 10%-12.5% of VLP fraction and cell lysate equivalent to 20 μg protein were subjected to SDS-PAGE with 4-15% Mini-Protean TGX precast gel (456-1086, Bio-Rad, CA). The results and

the uncropped images are shown in Supplementary Figs. 2 and 4 and the Source Data file, respectively.

## Electron microscopy observation of HIV-1 Gag VLP production on cell surface

The ultrastructures of Gag/Gag-EGFP-expressing HeLa cells were observed as reported previously[41] with slight modifications. Gag-transfected HeLa cells were grown on Aclar plastic sheets (Nisshin EM, Japan) in DMEM supplemented with 10% (v/v) FBS. Cells were fixed for 2 h at room temperature with 2% paraformaldehyde, 2% (v/v) glutaraldehyde, and 2 mM $CaCl_2$ in PHEM buffer (60 mM PIPES, 25 mM HEPES, 10 mM EGTA, and 2 mM $MgCl_2$ at pH 6.9), postfixed for 30 min with 2% osmium tetraoxide and 1.25% potassium ferrocyanide in PHEM buffer, followed by 2% osmium in PHEM buffer for another 30 min. The samples were then dehydrated in the graded series of ethanol, embedded in Araldite resin, and sectioned with an ultramicrotome (EM UC6; Leica). Ultrathin sections were stained with Uranyless EM stain (Electron Microscopy Sciences, Hatfield, PA) and lead citrate. Specimens were examined under the transmission electron microscope (H7500, Hitachi) with the help of the Institut des Neurosciences Cellulaires et Intégratives, Université de Strasbourg. The representative image is shown in Supplementary Fig. 3.

## Cell lysate prepared from Gag-derivatives expressing-cells

HeLa cells were seeded in 2 mL of DMEM/FBS/P/S at a density of $1.5 \times 10^5$ cells/mL in a 6-well plate (TPP, Switzerland). The next day HeLa cells were transfected with 1 μg of a 3:1 mixture of non-tagged and mTagBFP2- or mEos4b-tagged Gag plasmid using jetPRIME reagent according to the manufacturer's protocol. After 24 h of transfection, cells were washed twice with 0.8 mL of ice-cold PBS, collected by scraping in scraping buffer (25 mM Tris-Cl pH 7.4, 180 mM NaCl, and 1 mM EDTA) into 1.5 mL tube, and pelleted by centrifugation at $900 \times g$ for 5 min at 4 °C. The cell pellet was suspended in SHEI buffer, and then the suspension was sonicated briefly. The protein concentration of the lysate was determined by BCA protein assay kit (23228, Pierce, IL). Lysate equivalent to 20 μg of protein was subjected to SDS-PAGE with 4-15% Mini-Protean TGX precast gel. The representative images of membranes probed by Western are shown in Supplementary Figs. 5 and 6.

## Western blotting detection

The protein of interest on a PVDF membrane was probed using anti-HIV-1 p24 (ARP-3537, clone 183-H12-5C, lot# 130149, ATCC; dilution rate × 5000; https://www.hivreagentprogram.org/Catalog/HRPMonoclonalAntibodies/ARP-3537.aspx), anti-GFP (G10362, Invitrogen; × 3000; https://www.thermofisher.com/antibody/product/GFP-Antibody-Recombinant-Monoclonal/G10362), anti-γ-tubulin (T-5326, clone GTU-88, Sigma Aldrich; × 5000; https://www.sigmaaldrich.com/FR/fr/product/sigma/t5326), or anti-α-tubulin (T-9026, clone DM1A, Sigma Aldrich; × 2000; https://www.sigmaaldrich.com/FR/fr/product/sigma/t9026) as a primary antibody, and anti-mouse IgG horseradish peroxidase (HRP) conjugated (NA931V, lot# 17062552, GE Healthcare; × 5000) or anti-rabbit IgG HRP conjugated (NA934, lot# 17853438, GE Healthcare; × 5000) as a secondary antibody. For the detection, chemiluminescence produced by ECL Prime reagent (RPN2232, Cytiva) was imaged by LAS4000mini under the control of Image Reader LAS-4000 ver. 1.0. Uncropped images of probed membranes in Supplementary Figs. 2, 4, 5, and 6 are shown at the end of the Supplementary Information and in the Source Data file.

## Statistical analysis

All graphs were drawn in Microsoft Excel according to ref. 101 except for FLIM diagram plots, which were drawn by flimDiagRam. The significance tests of differences either between two sample means or among more than two sample means in three independent experiments were performed with paired two-tailed Student's *t*-test in

Microsoft Excel or repeated measures one-way ANOVA post-hoc Tukey test in GraphPad Prism, respectively. Pearson correlation coefficients were calculated in GraphPad Prism and Microsoft Excel to evaluate correlations between Gag-BFP2 intensity and lifetime $\tau_1$ or amplitude $\alpha_1$ in Fig. 6 and Supplementary Fig. 17. Fitting decay curves in FLIM-FRET analysis and the evaluation of the fitting ($\chi^2$) were performed with SPCImage (Becker & Hickl GmbH) software. All detailed statistics that are not described in Figures, Supplementary Figures, and Supplementary Tables are shown in the Source Data file[105].

## Reporting summary

Further information on research design is available in the Nature Portfolio Reporting Summary linked to this article.

## Data availability

All data generated in this study are shown in the Figures, Supplementary information, and Source Data file. Source data are provided with this paper.

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

## Acknowledgements

We are grateful to the NIH-AIDS Reagent Program (ARP) for providing us Gag and Gag mutant plasmids. We acknowledge the Imaging Center PIQ-QuESt (https://piq.unistra.fr/), a member of the national infrastructure France-Bioimaging supported by the French Research Agency (ANR-10-INBS-04). Y.M. is grateful to the Institut Universitaire de France (IUF) for support and for providing additional time to be dedicated to research. This work was supported by Agence Nationale de Recherche sur le Sida et les Hépatites Virale 18365 (T.K.), Agence Nationale de la Recherche A20R417C (T.K.), and RIKEN Glycolipidologue Program (T.K.).

## Author contributions

Conceptualization: N.T., Y.M., T.K. Methodology: N.T., P.D., J.G., L.R., Y.S., Y.M., T.K. Investigation: N.T., M.B.N., M.M., B.P. Visualization: N.T., M.B.N. Supervision: N.T., Y.S., Y.M., T.K. Writing—original draft: N.T., M.B.N., T.K. Writing—review & editing: N.T., M.B.N., M.M., B.P., P.D., J.G., L.R., Y.S., Y.M., T.K.

## Competing interests

The authors declare no competing interests.
