## [Peer Review File · Nature Communications]

Reviewers' Comments:

Reviewer #1:

Remarks to the Author:

Using their validated NT-Lys and D4 probes, the authors demonstrate that Gag proteins induce formation of denser AF647-NTLys domains possibly by reorganizing lipid domains. This result, while largely speculated by previous studies given the lipidomics data of lentivirus particles, more directly demonstrates that Gag is able to coalesce SM domains during the budding and assembly process. This result is followed by a predicted loss of bulk mobility of NT-Lys probes upon Gag expression, as assessed by FRAP. Further, the authors utilize a powerful FLIM-FRET approach to demonstrate molecular crowding of these lipid species at sites of Gag budding. This study then utilizes mutants of Gag to show that changes in bud curvature can influence the clustering of SM/chol domains.

This reviewer finds this study to be of high quality and it should find interest for both virologists and membrane biophysicists. However, before acceptance, the authors must address the following major and minor points to improve the study/manuscript.

Major comments:

1) The sizes of Gag-mEOS2 clusters seem larger than reported (~300 nm) in previous studies. The authors report the use of equimolar ratios of Gag-mEOS2 and untagged Gag. With the latter doping needed as use of 100% tagged Gag leads to assembly/budding defects (Jouvenet and Simon past work). However, many studies have utilized far more untagged / tagged Gag ratios in fluorescent imaging studies of Gag budding (1:3 and as high as 1:20) to prevent these artifacts. Can the authors show via standard thin-section EM that when introduced into cells in this identical ratio of Gag/Gag-mEOS2 (1:1) that there is not a budding defect? Similarly, at this ratio, is the amount of Gag released from cells (biochemically tested) result in a budding/release defect? Gag buds typically range in size centered around 150-165 nm. The larger area reported by the authors suggests that the mEOS2 tag is interfering with the membrane budding process and the Gag array remains flat on the plasma membrane in an artefactual way.

2) To further the previous point, mEOS2 is not truly monomeric and likely contributes to crosslinking of via mEOS2 dimers/tetramers when presented at a high effective concentration on the inner leaflet of the plasma membrane. Can the authors perform additional experiments with the truly monomeric mEOS3/4 derivatives that have now long been described? Do the sizes of Gag bud via super-resolution change significantly?

3) The authors should demonstrate that curvature is important for SM and Chol coalescence using the other reported curvature-deficient mutant EE-AA (75-76). The P99A mutant does seem to behave slightly differently from another curvature mutant as visualized by electron microscopy in those previous studies.

4) AlexaFluor647 has previously been shown to "contaminate" the EOS fluorescent channel by single molecule imaging. The authors should provide control (no Gag-mEOS2) data to show that this contamination of the EOS channel is not significant compared to their AF647-NT-Lys/Gag-mEOS2 dataset.

5) Why have the authors not shown PALM/STORM with Chol(D4) and Gag? This would improve the study significantly and make it possible to compare with FLIM-FRET data in SM/Chol.

Minor comments:

1) Can the authors show representative diffraction-limited FRET (A/D and A only excitation) images of cells analyzed by FRET-FLIM to demonstrate qualitative colocalization of these probes?

2) The term "Dot" is not commonly used. Typically in super-resolution microscopy, the term localization is used. The authors should be consistent with common nomenclature in the super-resolution field.

- 3) The authors use the term molecules when referring to the number of localizations that they observed for Gag-mEOS2. The authors should be careful as it has been shown that single mEOS2 molecules have reappearance events that are non-normally distributed.
- 4) The statement on page 4: "In contrast, only a low number of Gag-mEos2 domains did not associate with AF647-NT-Lys." This should be quantified and the percentage of Gag foci unlabeled by AF647-NT-Lys should be reported with error.
- 5) It is unclear whether the two cells analyzed for PALM/STORM analysis were performed on the same day/specimen coverslip or whether these are independent biological replicates performed on different days. Given the heterogeneity of both lipid domains and single molecule imaging, it would be best to have independent measurements and the differences in cluster measurements compared.
- 6) Shouldn't a statistical analysis be performed on the FLIM-FRET data to justify statistical differences between FLIM measurements under perturbation conditions such as +/- Gag expression.
- 7) What determines the oval/circular properties of the assembly site in the ROI? How was this controlled?
- 8) The demonstration that "Expression of Gag does not alter the gross lipid composition of HeLa cells" is an important control to support the author's results, but this brief paragraph at the end of the manuscript would likely be better commented as not being a factor earlier in the paper before presenting the bulk of data and this results paragraph moved into a supplementary figure/legend to explain why this is important to test.
- 9) Figure formatting, align images and place scale bars in the same region. Scale bar size text is unnecessary if stated in legend.
- 10) Fig. 2A. Placing a full cell image with these insets would be nice. Please add scale bars.
- 11) Fig. 3. A representative cell with FRAP area (before/after photobleaching or time series) to correspond with the plot would be helpful for the audience.
- 12) Add the complete single-molecule localization precisions for PALM/STORM to the supplement.
- 13) Different fonts are used for text throughout the figures/text.

Reviewer #2:

Remarks to the Author:

In this paper, Tomishige et al use advanced fluorescence microscopy to study the interaction of the HIV Gag protein with two plasma membrane lipids, namely sphingomyelin and cholesterol, during viral budding. HIV budding is a crucial process for virus biology, and its study can also help in the understanding of plasma membrane organisation. However, due to the difficulties of directly observing lipids, the exact budding mechanism is not fully understood yet. In 2019, two research articles addressed this question using live-cell fluorescence microscopy and fluorescent lipid analogues and proteins (Sengupta et al 2019 Nat Cell Biol and Favard et al 2019 Sci Adv). This work has the potential to complement the abovementioned studies, however it is my feeling that the conclusions are not supported by the data. While reading the paper I found several issues that in my opinion prevent publication of this work.

1. The paper would benefit from a more rigorous data treatment. In the Reporting Summary the authors write:

"The quantification of PALM/dSTORM images (Fig. 1, 2, and S1-S3) was replicated on two different cells. The measurements in confocal imaging (Fig. 1), FRAP (Fig. 3), FLIM-FRET (Fig. 4 and 5), and lipid content (Fig. 6) were repeated more than twice except that P99A and Δ L mutants in Fig. 5 were measured once."

Two cells are not enough. The authors should repeat the measurements done in two cells (Figs. 1-2) at least in three independent experiments, as well as the experiments with the P99A and ΔL mutants, which have been measured once. Variability in such experiments is relatively high, even in a cell-to-cell basis. This would increase the soundness of the presented results and conclusions. The number of cells measured on each independent experiment should also be stated.

I also suggest that comparison between the different tested conditions is presented as box plots showing independent experiments, so that experimental variability can be appreciated by the reader. This would apply to FRAP (Fig. 3) and FLIM results (Figs. 4-5). This recent work by Lord et al provides good examples of data presentation (<https://doi.org/10.1083/jcb.202001064>).

2. Citations.

I counted 25 citations to papers from the corresponding authors. Some examples:

-Page 2. "Binding of Gag to genomic RNA in the cytoplasm is accompanied by oligomerization of Gag (12, 13)." This is not the original research, but rather a citation to a review written by the authors and a later paper also from the authors. In other cases, like reference 52, the authors do cite the original paper, even if it is from 1908.

-Page 3. "In the PM of mammalian cells, lipids are asymmetrically distributed: (...) mainly located in the outer leaflet (18-20)." Self-citations of reviews by the authors, not original research.

-Page 16. "(...) with a 60 \times 1.2 NA water immersion objective operating in the descanned fluorescence collection mode as described in Azoulay et al. (78)." This self-citation is not necessary to describe a microscope.

-The discussion mostly consists of self-citations (I counted 9/16). Some provocative statements are made, for example in page 11 "These data show for the first time the presence of heterogeneous complexes or domains that may differ by the arrangement and/or packing of SM and Chol molecules using two different lipid probes." These strong statements should be discussed in the context of the vast literature on plasma membrane organisation.

Wrong or missing citations:

-Page 9. "Previous studies have shown that the Gag induced increase in membrane curvature favors the enrichment in membrane proteins preferring the Lo (liquid-ordered) lipid phase (35)." It is my understanding that this paper shows a curvature independent enrichment of the membrane protein CD59.

-The discussion on the lipid compositions results Fig. 6 should include a reference to Mücksch et al 2019 (reference 5 in the manuscript), where the lipidome of HIV infected and non-infected cells are compared.

3. At the beginning of the results section, Dots are defined as "a fluorescent signal derived from one fluorophore observed by superresolution." If dots refer to localisations, I suggest referring to them as such, as it is the standard in the field. If different localisations are combined in each "dot". How are they assigned to a single fluorophore?

4. The use of lipid-binding proteins is very valuable, and indeed a good alternative to fluorescent lipid analogues used in previous studies, that in many cases fail to reproduce the behaviour of the unlabelled lipids. However, lipid-binding proteins do not come without disadvantages, such as the fact that they only bind clusters of six sphingomyelin (SM) molecules (NT-Lys) or when the cholesterol concentration is greater than 40% (D4). I miss a discussion of the possible limitations of this approach. For example, unclustered lipids are not detectable, clusters might be induced by the proteins themselves, or binding to such proteins might also alter the behaviour of lipids.

In page 11 "This is in variance with the 10-fold higher diffusion coefficient reported for the lateral diffusion of SM (...). This discrepancy can be explained by the fact that ATTO647N-SM has been shown to distribute in Ld (liquid-disordered) domains due to the charged bulky fluorophore (38) (...)." This is most likely not correct. The diffusion of lipids in Ld environments in vesicles derived from plasma membranes is only slightly faster than in Lo domains, and as such a partitioning difference would not explain such big change. The most likely scenario is that EGFP-NT-Lys in complex with six sphingomyelin molecules will diffuse slower due to its high molecular weight (as compared to a lipid analogue), and additive frictional drags from each SM molecule (see Ziembra and Falke Chem Phys Lipids 2013). In short, six clustered SM molecules will always diffuse slower than a single molecule and will not recapitulate the behaviour of the unbound lipid.

5. Still, the results on cluster mobility are interesting. I suggest the authors to include D4 in their FRAP experiments to also study the changes in cholesterol mobility.

6. The SMLM experiments should include non-transfected cells as controls as in Fig. 3, i.e. how does SM organisation change upon Gag expression?

7. Page 5 "it can be inferred from Fig. 1D that 40% of Gag-mEos2 and 26% of AF647-NT-Lys

colocalized (Fig. 1D yellow region) under our experimental condition.” Does this mean that 60% of Gag-mEos2 does not colocalise with SM? If there is indeed an interaction with SM, how can the authors explain that most Gag does not show a colocalisation?

8. Have labelled protein such as Gag-mEos2, Gag-mTagBFP2, Alexa Fluor 647-NT-Lys or Alexa Fluor 546-SNAP-NT-Lys been published before? If not, validation data would be necessary. Do these Gag versions produce functional virus particles? Do Alexa Fluor tagged NT-Lys behave as the validated GFP version?

9. In line with the previous comment, do such high unlabelled:labelled Gag (1:1) ratios affect particle budding. I would imagine that steric hindrances caused by the fluorescent protein might reduce oligomerisation efficiency or curvature formation. The authors should validate some of their findings using lower ratios (1:8) or proof that 1:1 ratios yield functional viral particles.

10. I found the FLIM plots confusing. This representation has been recently developed by the authors (Godet and Mély 2019) and is not the standard for presenting FLIM data. Since the manuscript targets an interdisciplinary audience, further explanations would benefit the understanding of the paper. For example, the blue dashed lines that represent the average lifetime are not described.

11. Why do the authors use this alternative approach for FLIM? If I understood the method correctly, it is assumed that there are always two lifetime populations on every pixel. The authors claim that in the EGFP-NT-Lys sample (Fig. 4A) they resolve two lifetimes, 2.3 ns corresponding to EGFP and another of 2.2 ns corresponding to a pseudo-homo FRET population. However, in the supplementary validation, it seems that a single component function fits the data better. Moreover, the pseudo-homo FRET population seems to disappear in the presence of the acceptor. If the use of this alternative approach is justified, the authors should discuss it.

12. Throughout the manuscript, lipid domain fusion is claimed to be the mechanism for SM domain formation. This is a possible scenario, but no evidence for domain fusion is shown. Thus, I suggest that such claims are kept for the discussion section, as they could be considered speculative.

13. FRAP (Fig. 3) and FLIM (Figs. 4-5) experiments are performed upon transfection with fluorescent Gag versions. However, this is not exploited, as differences between Gag assembly sites and the rest of the membranes are not explored. Why are cells transfected with fluorescent Gag then? It would be interesting to know whether lifetime or diffusion changes are observed in and out of Gag assembly sites.

Moreover, one of the main advantages of such techniques is that they are combined with imaging. No images of such experiments are shown throughout the manuscript. I would encourage the authors to show images of these experiments.

14. Figs. S3D to F are missing from the Supplementary Information file made available to me. Figure S4G is also missing.

15. The authors claim that “The FLIM diagram plot [for the Gag-WM mutant, which is oligomerisation incompetent] was similar to that without Gag expression”. However, it also seems similar to that with Gag expression (Fig. 4C). The amplitude values shown in Table 1 are indeed closer to those without Gag, however the lifetime (1.27 ns) is the same as in the Gag expressing cells. Further statistical treatment of these data would help clarify this point. While the plots showing the amplitude/lifetime values for each pixel are appreciated to understand the raw data, comparison between average values for each cell and independent experiment would show the experimental variability and establish the significance of the observed differences.

16. The kernel density estimate function used to determine the number of populations in the FLIM data is not explained in the methods section. Have the authors validated that this method can distinguish four lifetime populations, as they do in Fig. 5 and Table 2 for the DA and WM samples (3 values for tau1 and 1 for tau2)? It also seems likely that the variability of the data just indicates that there is a broad range of FLIM values corresponding to a range of FRET distances between donor and acceptor, and not necessarily four distinct populations.

The abovementioned claim “These data show for the first time the presence of heterogeneous complexes or domains that may differ by the arrangement and/or packing of SM and Chol molecules using two different lipid probes.” Is solely based on this analysis, and as thus might be too speculative.

To further proof this, SMLM experiments studying the colocalisation between D4 and NT-Lys would be valuable.

17. Which mounting medium was used for SMLM experiments?

18. Why is two-photon FLIM used instead of one-photon FLIM? Does it offer any advantages in this case? The complex photophysics of two-photon excitation might hinder the interpretation of the

results.

19. The FLIM-FRET methods section misses some critical details. For example, what imaging medium was used? How long after transfection were the cells imaged? Is the imaging performed in the lower (coverslip) or upper plasma membrane? What is the pixel size? How long is the dwell time? Does the "binning off two" refer to two pixels?

20. The Data and materials availability statement claims: "All data are available in the main text or the supplementary materials". The raw data is not available in the figures, and since non-standard analysis methods are used, it would be valuable for the community if these data would be made available, even if it is upon request to the authors.

Reviewer #3:

Remarks to the Author:

The authors present a study into the mechanism behind the different lipid compositions of the HIV envelope and the plasma membrane of the producer cell from which it is derived. A particular focus is given to the outer leaflet lipid sphingomyelin, which is enriched in virions. The authors recently developed a sphingomyelin-specific probe, a non-toxic mutant of lysenin, and this paper is a demonstration of the utility of this probe particularly with respect to super-resolution fluorescence imaging. The manuscript is well-written, follows a logical flow, and the conclusions are well-supported by the data. Key findings of the paper are that the HIV Gag protein (responsible for virus budding) colocalised with sphingomyelin clusters and causes their reorganisation or coalescence into larger clusters despite Gag accumulating on the inner leaflet, and sphingomyelin in the outer leaflet. FRAP experiments suggest that Gag increases the 'immobile fraction' of sphingomyelin, restricting its diffusion when associated with Gag.

D4 (a cholesterol-specific probe) is used alongside lysenin to show the coalescence of sphingomyelin- and cholesterol-rich domains at Gag sites. The authors suggest that membrane curvature plays a role in the coalescence using Gag mutants: P99A which does self-assemble but does not cause membrane curvature and does not bud, and Gag-deltaL which does curve the membrane but is unable to recruit ESCRT machinery also required for budding. It would have been compelling to see the effect of these mutants on the previous experiments (colocalization, cluster size, and lateral diffusion) to exclude any changes in these properties due to mutation, leaving membrane curvature as the most-likely explanation for the sphingomyelin/cholesterol coalescence. It would also be useful to see (by EM) the nature of the curvature induced by these Gag mutants in these cells.

Importantly, expression of Gag does not change the global lipid composition in the cell. This result is an important control, and should probably be stated up front in the results section, rather than being added at the end. Without this result the other results would not be interpretable. It should also be performed for Gag mutants.

One important aspect of Gag that has not been mentioned in this manuscript is that the protein is N-terminally myristoylated. The myristoylation has, in the past, been suggested to target Gag to certain lipid domains. This manuscript would suggest that curvature is the main feature driving sphingomyelin colocalization and restructuring. It would be interesting to see what effect a myristoylation-deficient Gag mutant has on colocalization and sphingomyelin/cholesterol coalescence.

Minor comment: Fig 2B and S3F should be reworked for clarity. It isn't intuitively obvious the right graph is a zoom of the left, and it is not clear that the 50nm columns are truncated in the y-axis. Consider a split y-axis, or omitting the low value x-axis columns in the zoomed in data.

REVIEWER COMMENTS

Reviewer #1 (Remarks to the Author):

Using their validated NT-Lys and D4 probes, the authors demonstrate that Gag proteins induce formation of denser AF647-NTLys domains possibly by reorganizing lipid domains. This result, while largely speculated by previous studies given the lipidomics data of lentivirus particles, more directly demonstrates that Gag is able to coalesce SM domains during the budding and assembly process. This result is followed by a predicted loss of bulk mobility of NT-Lys probes upon Gag expression, as assessed by FRAP. Further, the authors utilize a powerful FLIM-FRET approach to demonstrate molecular crowding of these lipid species at sites of Gag budding. This study then utilizes mutants of Gag to show that changes in bud curvature can influence the clustering of SM/chol domains.

This reviewer finds this study to be of high quality and it should find interest for both virologists and membrane biophysicists. However, before acceptance, the authors must address the following major and minor points to improve the study/manuscript.

Major comments:

1) The sizes of Gag-mEOS2 clusters seem larger than reported (~300 nm) in previous studies. The authors report the use of equimolar ratios of Gag-mEOS2 and untagged Gag. With the latter doping needed as use of 100% tagged Gag leads to assembly/budding defects (Jouvenet and Simon past work). However, many studies have utilized far more untagged / tagged Gag ratios in fluorescent imaging studies of Gag budding (1:3 and as high as 1:20) to prevent these artifacts. Can the authors show via standard thin-section EM that when introduced into cells in this identical ratio of Gag/Gag-mEOS2 (1:1) that there is not a budding defect? Similarly, at this ratio, is the amount of Gag released from cells (biochemically tested) result in a budding/release defect? Gag buds typically range in size centered around 150-165 nm. The larger area reported by the authors suggests that the mEOS2 tag is interfering with the membrane budding process and the Gag array remains flat on the plasma membrane in an artefactual way.

We appreciate the reviewer for the valuable suggestions.

1. Concerning the Gag fusion protein, reviewer #1 cited the paper of Jouvenet and Simon (Jouvenet et al., 2008). In this paper, they used Gag-GFP in which GFP was fused to the C-terminus of Gag. In our case, fluorescent proteins were inserted between MA and CA domain of Gag protein (Gag-FP). A 1:1 mixture of this Gag-FP and non-tagged Gag showed the production of the virion with normal morphology and full wild-type infectivity (Muller et al., 2004). But the effect of this Gag-FP on VLP budding has not been examined. Although we already described the information on the FP insertion

in Methods of the previous manuscript, we newly added a detailed description about the Gag-FP construct in the revised text (page 4, line 127-138 in Results; page 17, line 609-611 in Methods).

2. Concerning the ratio of Gag/Gag-FP, as requested by the referees, we newly repeated all the experiments with a ratio of 3:1 for Gag/Gag-FP. To show the absence of the budding defects at this new ratio, we attached Supplementary Fig. 2 demonstrating the existence of Gag/Gag-EGFP (3:1) in VLP fraction and cell lysate. In this figure, we found that in either VLP or lysate, the total content of Gag (Gag + Gag-EGFP) is similar between Gag alone and Gag/Gag-EGFP (3:1) -transfected samples, indicating that this ratio (3:1) of Gag/Gag-FP did not affect VLP production. The morphology and production of VLP at the HeLa cell surface also seemed to be not affected (Fig. S3). This is described in the new text (page 4, line 133-138).

3. We replaced Gag-mEos2 with Gag-mEos4b in our new PALM/dSTORM experiments to avoid dimerization problems as suggested by reviewers #1 and #2. We estimated the size of Gag-mEos4b and AF647-NT-Lys domains and mentioned it in the revised text (page 6, line 215-220, and page 6-7, line 228-230). The diameter estimation of Gag WT-mEos4b domain in a whole cell area by SR-Tesseler program showed 179 ± 13 nm (mean \pm SEM).

In the literature, HIV-1 budding sites and extracellular particles have an average diameter of 120-140 nm with significant size variability as determined by cryo-electron microscopy (Briggs et al., 2009; Carlson et al., 2008; Larson et al., 2005; Wright et al., 2007). When estimating the diameter of Gag domains (arrows in Supplementary Fig. 8) that were isolated from other large domains (arrowheads in Supplementary Fig. 8) by setting criteria ($85 \text{ nm} < \text{diameter} < 170 \text{ nm}$, localization number > 50 , and circularity# > 0.8) similar to the previous SMLM study (Lehmann et al., 2011), these isolated Gag domains showed a mean diameter of 119 ± 3 nm (mean \pm SEM). This value is consistent with the values (117 ± 45 nm, mean \pm SD) reported in Lehmann et al., 2011 and the above reported particle diameters. This indicates that our new PALM/dSTORM experiment is properly performed.

#: Circularity = $(4\pi \cdot \text{area}) / (\text{perimeter})^2$, where a circle shows the circularity equal to 1.

Jouvenet, N., P.D. Bieniasz, and S.M. Simon. 2008. Imaging the biogenesis of individual HIV-1 virions in live cells. *Nature* **454**, 236-240.

Muller, B., J. Daecke, O.T. Fackler, M.T. Dittmar, H. Zentgraf, and H.G. Krausslich. 2004. Construction and characterization of a fluorescently labeled infectious human immunodeficiency virus type 1 derivative. *J. Virol.* **78**, 10803-10813.

Briggs, J.A., J.D. Riches, B. Glass, V. Bartonova, G. Zanetti, and H.G. Krausslich. 2009. Structure and assembly of immature HIV. *Proc. Natl. Acad. Sci. U. S. A.* **106**, 11090-11095.

Carlson, L.A., J.A. Briggs, B. Glass, J.D. Riches, M.N. Simon, M.C. Johnson, B. Muller, K. Grunewald, and H.G. Krausslich. 2008. Three-dimensional analysis of budding sites and released virus suggests a revised model for HIV-1 morphogenesis. *Cell host & microbe* **4**, 592-599.

Larson, D.R., M.C. Johnson, W.W. Webb, and V.M. Vogt. 2005. Visualization of retrovirus budding with correlated light and electron microscopy. *Proc. Natl. Acad. Sci. U.S.A.* **102**, 15453-15458.

Wright, E.R., J.B. Schooler, H.J. Ding, C. Kieffer, C. Fillmore, W.I. Sundquist, and G.J. Jensen. 2007. Electron cryotomography of immature HIV-1 virions reveals the structure of the CA and SP1 Gag shells. *EMBO J.* **26**, 2218-2226.

Lehmann, M., S. Rocha, B. Mangeat, F. Blanchet, I.H. Uji, J. Hofkens, and V. Piguet. 2011. Quantitative multicolor super-resolution microscopy reveals tetherin HIV-1 interaction. *PLoS Pathog.* **7**, e1002456.

2) To further the previous point, mEOS2 is not truly monomeric and likely contributes to crosslinking of via mEOS2 dimers/tetramers when presented at a high effective concentration on the inner leaflet of the plasma membrane. Can the authors perform additional experiments with the truly monomeric mEOS3/4 derivatives that have now long been described? Do the sizes of Gag bud via super-resolution change significantly?

We thank the reviewer for this critical comment. According to the referee's suggestion, we newly performed PALM/dSTORM imaging with Gag-mEos4b. PM labeling pattern by Gag-mEos4b and its domain sizes were different from those by Gag-mEos2 in the super-resolution microscopy, as mentioned in the answer to the previous comment. The results are also shown in the revised manuscript (Fig. 2, 3, and 7).

3) The authors should demonstrate that curvature is important for SM and Chol coalescence using the other reported curvature-deficient mutant EE-AA (75-76). The P99A mutant does seem to behave slightly differently from another curvature mutant as visualized by electron microscopy in those previous studies.

As suggested by the referee, we newly performed FLIM-FRET and PALM/dSTORM experiments using the Gag-EE mutant (here, we call the E75A E76A mutant EE mutant) in addition to Gag-P99A mutant. EE mutant showed essentially the same results as the P99A mutant in our experiments. These results were included in the revised manuscript (page 10-12, line 376-379, 385-388, 393-395, 398-399, 409-410; Fig. 6 and 7).

4) AlexaFluor647 has previously been shown to "contaminate" the EOS fluorescent channel by single molecule imaging. The authors should provide control (no Gag-mEOS2) data to show that this contamination of the EOS channel is not significant compared to their AF647-NT-Lys/Gag-mEOS2 dataset.

As the reviewer pointed out, AF647 has been reported to photoconvert to blue-shifted molecules by strong 561 or 642 nm laser irradiation in super-resolution fluorescence imaging (Dirix et al., 2018; Helmerich et al., 2021). This “photoblueing” can be suppressed by using an appropriate medium containing reducing agents and scavengers of reactive oxygen species (Dirix et al., 2018; Helmerich et al., 2021). To check the effect of the photoblueing on our PALM/dSTORM imaging and the effect of the above medium on our system, we acquired images of cells labeled with AF647-lipid probe alone and with both AF647-lipid probe and Gag/Gag-mEos4b in the imaging buffer containing a reducing agent (100 mM MEA, beta-mercaptoethylamine) and oxygen scavenger system (GLOX, 56 mg/mL of glucose oxidase and 17 mg/mL of catalase in 10 mM Tris-Cl pH8.0 and 50 mM NaCl). “Contaminated” Alexa Fluor 647 signals in the mEos channel were negligible in this condition (below, Reviewer only Fig. 1 and 2). The details of this acquisition condition were described in Methods (page 19, line 701-713 in “PALM/dSTORM”).

In Reviewer only Fig. 1 and 2, panels a and b show representative reconstituted images of the merged channel and mEos4b-channel. These images clearly showed that “contaminated” AF647 localizations were much less detected in the mEos4b channel (panels c and d) and did not contribute to making a false Gag-mEos4b image in the absence of Gag expression (a, lower panel). In addition, when calculating the colocalization coefficient, Mander’s overlap coefficient (MOC), between contaminant AF647 signals in the mEos4b channel and AF647-NT-Lys (or AF647-D4) in the AF647 channel, MOC values of AF647 to mEos4b channel were low (panel e), indicating that the contribution of the contaminant AF647 signals to the colocalization analysis is negligible. We also showed MOC values of mEos4b to AF647 channel, which were relatively high (panel f). Because the MOC values are calculated as a proportion of the overlapped (colocalized) to the whole fraction, MOC values tend to be affected in cases where the total number of detections in the mEos4b channel was very low, and some of them were overlapped with/close to AF647 localizations (panel a, c, and d). However, placing appropriate controls should prevent misleading interpretations.

Dirix, L., K. Kennes, E. Fron, Z. Debyser, M. van der Auweraer, J. Hofkens, and S. Rocha.

Photoconversion of far-red organic dyes: Implications for multicolor super-resolution imaging. *ChemPhotoChem*. **2**, 433-441 (2018).

Helmerich, D.A., G. Beliu, S.S. Matikonda, M.J. Schnermann, and M. Sauer. Photoblueing of organic dyes can cause artifacts in super-resolution microscopy. *Nat. Methods* **18**, 253-257 (2021).

5) Why have the authors not shown PALM/STORM with Chol(D4) and Gag? This would improve the study significantly and make it possible to compare with FLIM-FRET data in SM/Chol.

We thank the reviewer for the comment. As requested by the referee, we newly carried out PALM/dSTORM with AF647-D4 and Gag. We found similarities as well as differences with AF647-NT-Lys/Gag in the colocalization and domain analyses. These findings contributed to our

better understanding of the reorganization of lipid domains in the PM. These results are shown in Fig. 2 and 3 of the revised version.

Minor comments:

1) Can the authors show representative diffraction-limited FRET (A/D and A only excitation) images of cells analyzed by FRET-FLIM to demonstrate qualitative colocalization of these probes?

Since we repeated FLIM-FRET experiments with a new ratio of Gag/Gag-mTagBFP2 (3:1), we took fluorescent images of EGFP-NT-Lys and Gag-mTagBFP2. However, we could not address the qualitative colocalization between the donor and acceptor due to the lack of a third detector for acceptor images (mCherry-D4) in our microscope setup in addition to the above two channels. Instead, we found a correlation between EGFP lifetime and Gag-mTagBFP2 intensity. Interestingly, this correlation was observed in cells expressing Gag WT and Δ L but not other mutants (except for a slight correlation between amplitude α_1 and mTagBFP2 intensity in EE mutant). We showed these new data in Fig. 6d, 6e, and 6f. We also added cell images (Gag WT/Gag WT-mTagBFP2- or Gag-WM/Gag-WM-mTagBFP2-expressing cells labeled with donor (EGFP-NT-Lys) and acceptor (mCherry-D4)) acquired under the same condition as the FLIM-FRET experiment in Fig. 2a.

2) The term “Dot” is not commonly used. Typically in super-resolution microscopy, the term localization is used. The authors should be consistent with common nomenclature in the super-resolution field.

As suggested, we replaced “dot” by “localization” throughout the revised manuscript.

3) The authors use the term molecules when referring to the number of localizations that they observed for Gag-mEOS2. The authors should be careful as it has been shown that single mEOS2 molecules have reappearance events that are non-normally distributed.

We would like to thank the reviewer for this comment. We carefully revised the data presentation and interpretation and remove the term “molecule” to refer to the number of localizations in the revised manuscript.

4) The statement on page 4: “In contrast, only a low number of Gag-mEos2 domains did not associate with AF647-NT-Lys.” This should be quantified and the percentage of Gag foci unlabeled by AF647-NT-Lys should be reported with error.

We cannot directly answer this comment because we repeated all the experiments and completely changed the analysis and data presentation in the new manuscript. Instead, we compared domain

sizes in cells without and with Gag expression in Fig. 3 as requested by referee#2 (comment#6) as well and compared colocalization between AF647-NT-Lys and Gag WT-mEos4b or Gag-WM-mEos4b in Fig. 2.

5) It is unclear whether the two cells analyzed for PALM/STORM analysis were performed on the same day/specimen coverslip or whether these are independent biological replicates performed on different days. Given the heterogeneity of both lipid domains and single molecule imaging, it would be best to have independent measurements and the differences in cluster measurements compared.

As mentioned above, according to the referees' requests, we replaced the previous results with new ones from three independent experiments. We analyzed data from 4-6 cells/sample in each experiment and treated them with statistical tests. We included these results in Fig. 2, 3, and 7.

6) Shouldn't a statistical analysis be performed on the FLIM-FRET data to justify statistical differences between FLIM measurements under perturbation conditions such as +/- Gag expression.

As requested, we also performed three independent measurements in the FLIM-FRET experiment under the condition without and with Gag expression. These new data with statistical tests are shown in Fig. 5 and 6.

7) What determines the oval/circular properties of the assembly site in the ROI? How was this controlled?

In the previous version, we selected domains with oval/circular looking. In the revised manuscript, we newly repeated the experiments to compare domain sizes between cells expressing a vector and Gag and analyzed the data differently from the previous one. We thus completely changed the text (page 6-7, line 207-243, "Gag association enlarges AF647-NT-Lys-positive lipid domains") and data presentation (Fig. 3) in the new manuscript.

8) The demonstration that "Expression of Gag does not alter the gross lipid composition of HeLa cells" is an important control to support the author's results, but this brief paragraph at the end of the manuscript would likely be better commented as not being a factor earlier in the paper before presenting the bulk of data and this results paragraph moved into a supplementary figure/legend to explain why this is important to test.

Reviewer#3 also suggested showing this data first. We thus moved this paragraph and data to the beginning of the manuscript (page 4, line 140-151, "Expression of Gag does not alter the gross lipid composition of HeLa cells"; Fig. 1).

9) Figure formatting, align images and place scale bars in the same region. Scale bar size text is unnecessary if stated in legend.

We formatted all figures according to the publisher's instructions. We removed unnecessary texts from scale bars and added them to the legends.

10) Fig. 2A. Placing a full cell image with these insets would be nice. Please add scale bars.

As mentioned earlier, we completely changed the data presentation. We added whole cell images obtained in the new PALM/dSTORM imaging in Supplementary Fig. 6, 7, and 15 and in dSTORM imaging in Supplementary Fig. 14. We also added scale bars in all images of the revised manuscript.

11) Fig. 3. A representative cell with FRAP area (before/after photobleaching or time series) to correspond with the plot would be helpful for the audience.

We added representative images of FRAP area in a cell over chasing time in Fig. 4a, b, e, and f.

12) Add the complete single-molecule localization precisions for PALM/STORM to the supplement.

The localization precisions corresponding to our new super-resolution imaging are now given in the Supplementary Text ("Localization uncertainty of fluorescent probes in the super-resolution microscopy").

13) Different fonts are used for text throughout the figures/text.

We formatted the fonts in all the figures/text according to the publisher's instructions.

Reviewer #2 (Remarks to the Author):

In this paper, Tomishige et al use advanced fluorescence microscopy to study the interaction of the HIV Gag protein with two plasma membrane lipids, namely sphingomyelin and cholesterol, during viral budding. HIV budding is a crucial process for virus biology, and its study can also help in the understanding of plasma membrane organisation. However, due to the difficulties of directly observing lipids, the exact budding mechanism is not fully understood yet. In 2019, two research articles addressed this question using live-cell fluorescence microscopy and fluorescent lipid analogues and proteins (Sengupta et al 2019 Nat Cell Biol and Favard et al 2019 Sci Adv). This work has the potential to complement the abovementioned studies, however it is my feeling

that the conclusions are not supported by the data. While reading the paper I found several issues that in my opinion prevent publication of this work.

1. The paper would benefit from a more rigorous data treatment. In the Reporting Summary the authors write:

“The quantification of PALM/dSTORM images (Fig. 1, 2, and S1-S3) was replicated on two different cells. The measurements in confocal imaging (Fig. 1), FRAP (Fig. 3), FLIM-FRET (Fig. 4 and 5), and lipid content (Fig. 6) were repeated more than twice except that P99A and Δ L mutants in Fig. 5 were measured once.”

Two cells are not enough. The authors should repeat the measurements done in two cells (Figs. 1-2) at least in three independent experiments, as well as the experiments with the P99A and Δ L mutants, which have been measured once. Variability in such experiments is relatively high, even in a cell-to-cell basis. This would increase the soundness of the presented results and conclusions. The number of cells measured on each independent experiment should also be stated.

I also suggest that comparison between the different tested conditions is presented as box plots showing independent experiments, so that experimental variability can be appreciated by the reader. This would apply to FRAP (Fig. 3) and FLIM results (Figs. 4-5). This recent work by Lord et al provides good examples of data presentation (<https://doi.org/10.1083/jcb.202001064>).

As suggested by the reviewer, we newly repeated all these experiments independently three times. In PALM/dSTORM, FRAP, and FLIM-FRET, 4-6 cells, 7-11 cells, and 9 or 10 cells, respectively, were analyzed in each experiment. The detailed sample numbers over three experiments are following: in PALM/dSTORM, n = 16 (P99A, EE, and WM), 17 (vec and Δ L), and 18 (WT) for AF647-NT-Lys and Gag-mEos4b combination, n = 15 (vec), 17 (WT), and 16 (WM) for AF647-D4 and Gag-mEos4b combination, and n = 18 (vec) and 15 (WT) for AF647-NT-Lys and JF549-D4 combination; in FRAP, n = 29 (vec) and 28 (Gag) for EGFP-NT-Lys, and 30 (vec), and 25 (Gag) for EGFP-D4; in FLIM-FRET, n = 30 (vec, WT, and WM) and 29 (DA) for EGFP-NT-Lys and mCherry-NT-Lys combination, and n = 30 (vec, DA, WT, Δ L, P99A, and WM) and 29 (EE) for EGFP-NT-Lys and mCherry-D4 combination. We also described these sample numbers (n) in the legends of figures and the Reporting Summary. We also changed the data presentation of all microscopy data according to Lord et al. to show differences and statistics.

2. Citations.

I counted 25 citations to papers from the corresponding authors. Some examples:

-Page 2. “Binding of Gag to genomic RNA in the cytoplasm is accompanied by oligomerization of Gag (12, 13).” This is not the original research, but rather a citation to a review written by the authors and a later paper also from the authors. In other cases, like reference 52, the authors do cite the original paper, even if it is from 1908.

In the revised manuscript, we cited the original papers (Hubner et al., 2007; Kutluay and Bieniasz, 2010; El meshri et al., 2015; Hendrix et al., 2015) instead of the review (page 2, line 69-70). We also tried to cite original articles throughout the manuscript.

14. Hubner, W., P. Chen, A. Del Portillo, Y. Liu, R.E. Gordon, and B.K. Chen. Sequence of human immunodeficiency virus type 1 (HIV-1) Gag localization and oligomerization monitored with live confocal imaging of a replication-competent, fluorescently tagged HIV-1. *J. Virol.* **81**, 12596-12607 (2007).
15. Kutluay, S.B., and P.D. Bieniasz. Analysis of the initiating events in HIV-1 particle assembly and genome packaging. *PLoS Pathog.* **6**, e1001200 (2010).
16. El Meshri, S. E., Dujardin, D., Godet, J., Richert, L., Boudier, C., Darlix, J. L., Didier, P., Mely, Y., and de Rocquigny, H. Role of the nucleocapsid domain in HIV-1 Gag oligomerization and trafficking to the plasma membrane: a fluorescence lifetime imaging microscopy investigation. *J. Mol. Biol.* **427**, 1480-1494 (2015).
17. Hendrix, J., Baumgartel, V., Schrimpf, W., Ivanchenko, S., Digman, M. A., Gratton, E., Krausslich, H. G., Muller, B., and Lamb, D. C. Live-cell observation of cytosolic HIV-1 assembly onset reveals RNA-interacting Gag oligomers. *J. Cell Biol.* **210**, 629-646 (2015).

-Page 3. "In the PM of mammalian cells, lipids are asymmetrically distributed: (...) mainly located in the outer leaflet (18-20)." Self-citations of reviews by the authors, not original research.

In the revised version, we now cited the original research (Bretscher, 1972; Verkleij et al., 1973; Murate et al., 2015) (page 2, line 74-76).

24. Bretscher, M.S. Asymmetrical lipid bilayer structure for biological membranes. *Nat. New Biol.* **236**, 11-12 (1972).
25. Verkleij, A.J., R.F. Zwaal, B. Roelofsen, P. Comfurius, D. Kastelijn, and L.L. van Deenen. The asymmetric distribution of phospholipids in the human red cell membrane. A combined study using phospholipases and freeze-etch electron microscopy. *Biochim. Biophys. Acta* **323**, 178-193 (1973).
26. Murate, M., M. Abe, K. Kasahara, K. Iwabuchi, M. Umeda, and T. Kobayashi. Transbilayer distribution of lipids at nano scale. *J. Cell Sci.* **128**, 1627-1638 (2015).

-Page 16. "(...) with a 60× 1.2 NA water immersion objective operating in the descanned fluorescence collection mode as described in Azoulay et al. (78)." This self-citation is not necessary to describe a microscope.

Because our microscope system was assembled by ourselves, this explanation is necessary to properly describe our set-up. We thus prefer to keep this reference.

-The discussion mostly consists of self-citations (I counted 9/16). Some provocative statements are made, for example in page 11 “These data show for the first time the presence of heterogeneous complexes or domains that may differ by the arrangement and/or packing of SM and Chol molecules using two different lipid probes.”. These strong statements should be discussed in the context of the vast literature on plasma membrane organisation.

As requested, we now reduced the ratio of self-citations (10/29) by increasing others’ citations. However, citing our previous works is inevitable to discuss results obtained by lipid probes. We also toned down the above statement (page 14, line 519-521) and other statements.

Wrong or missing citations:

-Page 9. “Previous studies have shown that the Gag induced increase in membrane curvature favors the enrichment in membrane proteins preferring the Lo (liquid-ordered) lipid phase (35).”. It is my understanding that this paper shows a curvature independent enrichment of the membrane protein CD59.

Because we changed the text in the new manuscript, we removed this description.

-The discussion on the lipid compositions results Fig. 6 should include a reference to Mücksch et al 2019 (reference 5 in the manuscript), where the lipidome of HIV infected and non-infected cells are compared.

As requested, we added the reference of Mücksch et al 2019 (reference#5 in the revised manuscript) to refer to the lipidomics study of comparison of HIV-infected and non-infected cells (page 2, line 57-59).

3. At the beginning of the results section, Dots are defined as “a fluorescent signal derived from one fluorophore observed by superresolution.” If dots refer to localisations, I suggest referring to them as such, as it is the standard in the field. If different localisations are combined in each “dot”. How are they assigned to a single fluorophore?

The referee is right. Accordingly, we changed the term “dot” to “localization” throughout the revised manuscript.

4. The use of lipid-binding proteins is very valuable, and indeed a good alternative to fluorescent lipid analogues used in previous studies, that in many cases fail to reproduce the behaviour of the

unlabelled lipids. However, lipid-binding proteins do not come without disadvantages, such as the fact that they only bind clusters of six sphingomyelin (SM) molecules (NT-Lys) or when the cholesterol concentration is greater than 40% (D4). I miss a discussion of the possible limitations of this approach. For example, unclustered lipids are not detectable, clusters might be induced by the proteins themselves, or binding to such proteins might also alter the behaviour of lipids.

We added the detailed potential and limitation of lipid-binding proteins in the discussion (page 12, line 438-445; page 13, line 491-493).

In page 11 “This is in variance with the 10-fold higher diffusion coefficient reported for the lateral diffusion of SM (...). This discrepancy can be explained by the fact that ATTO647N-SM has been shown to distribute in Ld (liquid-disordered) domains due to the charged bulky fluorophore (38) (...).” This is most likely not correct. The diffusion of lipids in Ld environments in vesicles derived from plasma membranes is only slightly faster than in Lo domains, and as such a partitioning difference would not explain such big change. The most likely scenario is that EGFP-NT-Lys in complex with six sphingomyelin molecules will diffuse slower due to its high molecular weight (as compared to a lipid analogue), and additive frictional drags from each SM molecule (see Ziemba and Falke *Chem Phys Lipids* 2013). In short, six clustered SM molecules will always diffuse slower than a single molecule and will not recapitulate the behaviour of the unbound lipid.

We appreciate the referee for this comment and the information. We agree that the difference in the diffusions between the previous report and our results may be due to the difference in targets of the two probes. As suggested, we changed the sentence to “This discrepancy may be explained by the difference in these probes. ATTO647-SM is a lipid analog that has a charged bulky fluorophore attached to the acyl chain (Klymchenko and Kreder, 2014; Mobarak et al., 2018), whereas NT-Lys recognizes a SM cluster consisting of 5-6 molecules of SM (Ishitsuka et al., 2004; Ziemba and Falke, 2013)” (page 13, line 491-493).

Klymchenko, A.S., and R. Kreder. Fluorescent probes for lipid rafts: from model membranes to living cells. *Chem. Biol.* **21**, 97-113 (2014).

Mobarak, E., M. Javanainen, W. Kulig, A. Honigmann, E. Sezgin, N. Aho, C. Eggeling, T. Rog, and I. Vattulainen. How to minimize dye-induced perturbations while studying biomembrane structure and dynamics: PEG linkers as a rational alternative. *Biochim. Biophys. Acta* **1860**, 2436-2445 (2018).

Ishitsuka, R., A. Yamaji-Hasegawa, A. Makino, Y. Hirabayashi, and T. Kobayashi. A lipid-specific toxin reveals heterogeneity of sphingomyelin-containing membranes. *Biophys. J.* **86**, 296-307 (2004).

Ziemba, B.P., and J.J. Falke. Lateral diffusion of peripheral membrane proteins on supported lipid bilayers is controlled by the additive frictional drags of (1) bound lipids and (2) protein

domains penetrating into the bilayer hydrocarbon core. *Chem. Phys. Lipids.* **172-173**, 67-77 (2013).

5. Still, the results on cluster mobility are interesting. I suggest the authors to include D4 in their FRAP experiments to also study the changes in cholesterol mobility.

We thank the reviewer for this interesting suggestion. As requested, we newly added FRAP results of EGFP-D4 in the presence or absence of Gag expression. We found that the expression of Gag significantly decreased the mobile fraction of SM-rich domains, and Chol-rich domains were intrinsically immobile even in the absence of Gag. We included these results in Results section “Expression of Gag restricts the lateral diffusion of cell surface SM-rich domain” (page 7-8, line 251-257, 265-268) and in Fig. 4.

6. The SMLM experiments should include non-transfected cells as controls as in Fig. 3, i.e. how does SM organisation change upon Gag expression?

As requested, we added new PALM/dSTORM data including non-transfected cells as controls in Fig. 2 and 3, and Supplementary Fig. 6, 7, and 14. These new data helped us to show that Gag expression significantly increased the mean size of sphingomyelin-rich domains labeled with Alexa Fluor 647-SNAP-NT-Lys (AF647-NT-Lys) but not cholesterol-rich domains labeled with Alexa Fluor 647-D4 (AF647-D4) (Fig. 3 and “Gag association enlarges AF647-positive SM-rich lipid domains” in Results (page 6-7, line 207-243)), and that Gag expression significantly increased the colocalization between AF647-NT-Lys labeled SM-rich-domains and AF647-D4-labeled cholesterol-rich domains (Fig.6 and “Gag multimerization induces SM-rich and Chol-rich domains to be in close proximity” in Results (page 9, line 317-371)).

7. Page 5 “it can be inferred from Fig. 1D that 40% of Gag-mEos2 and 26% of AF647-NT-Lys colocalized (Fig. 1D yellow region) under our experimental condition.” Does this mean that 60% of Gag-mEos2 does not colocalise with SM? If there is indeed an interaction with SM, how can the authors explain that most Gag does not show a colocalisation?

The referee is right. If 40% of Gag-mEos2 (55% of Gag-mEos4b in the revised manuscript) colocalizes with SM, this means that 60% (45% in the revised manuscript) does not colocalize with NT-Lys-labeled SM. This can be explained in part by the fact that NT-Lys only binds to clusters of 5-6 SM molecules (Ishitsuka et al., 2004). As it has been previously reported, there is a SM pool in the cell membranes to which NT-Lys does not bind (Makino et al., 2015; Yachi et al., 2012). Therefore, we cannot detect Gag molecules that colocalize with SM in such SM pool. The limitation of NT-Lys to detect SM is written in Discussion (page 12, line 438-441).

- Ishitsuka, R., A. Yamaji-Hasegawa, A. Makino, Y. Hirabayashi, and T. Kobayashi. A lipid-specific toxin reveals heterogeneity of sphingomyelin-containing membranes. *Biophys. J.* **86**, 296-307 (2004).
- Makino, A., M. Abe, M. Murate, T. Inaba, N. Yilmaz, F. Hullin-Matsuda, T. Kishimoto, N.L. Schieber, T. Taguchi, H. Arai, G. Anderluh, R.G. Parton, and T. Kobayashi. Visualization of the heterogeneous membrane distribution of sphingomyelin associated with cytokinesis, cell polarity, and sphingolipidosis. *FASEB. J.* **29**, 477-493 (2015).
- Yachi, R., Y. Uchida, B.H. Balakrishna, G. Anderluh, T. Kobayashi, T. Taguchi, and H. Arai. Subcellular localization of sphingomyelin revealed by two toxin-based probes in mammalian cells. *Genes Cells* **17**, 720-727 (2012).

8. Have labelled protein such as Gag-mEos2, Gag-mTagBFP2, Alexa Fluor 647-NT-Lys or Alexa Fluor 546-SNAP-NT-Lys been published before? If not, validation data would be necessary. Do these Gag versions produce functional virus particles? Do Alexa Fluor tagged NT-Lys behave as the validated GFP version?

To repeat the experiments according to referees' suggestions, we changed some Gag constructs and labeled proteins. We newly introduced Gag-mEos4b, Alexa Fluor 647-SNAP-NT-Lys, Alexa Fluor 647-SNAP-D4, Janelia Fluor 549-SNAP-D4, as well as Gag-mTagBFP2 in this study. We added validation data of these new tools in Supplementary Fig. 1-5.

9. In line with the previous comment, do such high unlabelled:labelled Gag (1:1) ratios affect particle budding. I would imagine that steric hindrances caused by the fluorescent protein might reduce oligomerisation efficiency or curvature formation. The authors should validate some of their findings using lower ratios (1:8) or proof that 1:1 ratios yield functional viral particles.

As mentioned above, we changed some tools and conditions to accomplish new experiments. To perform those experiments, we chose a ratio of 1:3 for fluorescent protein (FP)-/non-tagged Gag because of limiting the steric hindrance possibly caused by Gag-FP. At the same time, this ratio allowed us to find cells expressing Gag more easily in various microscopy used in this study because the fluorescence was bright enough. Western blotting with anti-p24 antibody (Supplementary Fig. 2) showed almost the same total expression level of Gag protein (non-tag Gag + Gag-EGFP) in either lysate or VLP fraction between cells expressing only non-tagged Gag and non-tagged Gag/Gag-EGFP (3:1). We also added a picture of VLP budding from HeLa cell surface by transmission electron microscopy in Supplementary Fig. 3. These results supported that FP tagging of Gag between MA and CA domains did not affect the Gag expression and VLP formation at this ratio.

10. I found the FLIM plots confusing. This representation has been recently developed by the authors (Godet and Mély 2019) and is not the standard for presenting FLIM data. Since the manuscript targets an interdisciplinary audience, further explanations would benefit the understanding of the paper. For example, the blue dashed lines that represent the average lifetime are not described.

We moved the FLIM plots in Supplementary Fig. 12 and newly added data presentations following Lord et al. (Lord et al., 2020), as suggested by reviewer#1, in Fig. 5 and 6. We also added detailed explanations for readers about features on the FLIM plot in the legend of Supplementary Fig.12, which was pointed out by reviewer#2.

Lord, S. J., Velle, K. B., Mullins, R. D., and Fritz-Laylin, L. K. SuperPlots: Communicating reproducibility and variability in cell biology. *J. Cell Biol.* **219**, e202001064 (2020).

11. Why do the authors use this alternative approach for FLIM? If I understood the method correctly, it is assumed that there are always two lifetime populations on every pixel.

The authors claim that in the EGFP-NT-Lys sample (Fig. 4A) they resolve two lifetimes, 2.3 ns corresponding to EGFP and another of 2.2 ns corresponding to a pseudo-homo FRET population. However, in the supplementary validation, it seems that a single component function fits the data better. Moreover, the pseudo-homo FRET population seems to disappear in the presence of the acceptor. If the use of this alternative approach is justified, the authors should discuss it.

As mentioned above, the FLIM plot has the unique possibility to determine the distribution of the FRET parameters for the FRET species (short-lived lifetime and associated amplitude). This information was found in our case to well discriminate the behavior of the different Gag mutants (Supplementary Fig. 12). Concerning the fits, the single-component function fits better the data only in the case of the donor alone sample (Supplementary Text). In all other cases, a double component fit was shown to provide better chi square values. In the text, we used the one-component fit only in Fig 5a and 6a to show the existence of FRET by comparing donor alone sample (D) and donor + acceptor sample (DA). All other FLIM data have been analyzed using a two-component function. We also realize that the pseudo-FRET explanation provided to explain the 2.2 ns lifetime in the two-component analysis of the EGFP-NT-Lys alone was very confusing. We treated the donor alone only with one component in the revised version and removed the pseudoFRET explanation in the revised manuscript.

12. Throughout the manuscript, lipid domain fusion is claimed to be the mechanism for SM domain formation. This is a possible scenario, but no evidence for domain fusion is shown. Thus, I suggest that such claims are kept for the discussion section, as they could be considered speculative.

We followed this suggestion. We modified the “fusion” in the result section throughout to the expression that represents well what we found in the experiments.

13. FRAP (Fig. 3) and FLIM (Figs. 4-5) experiments are performed upon transfection with fluorescent Gag versions. However, this is not exploited, as differences between Gag assembly sites and the rest of the membranes are not explored. Why are cells transfected with fluorescent Gag then? It would be interesting to know whether lifetime or diffusion changes are observed in and out of Gag assembly sites.

We thank the referee for this excellent remark. It is true that in our previous manuscript, we mainly used the fluorescent Gag proteins to select cells on which we investigated the membrane reorganization. In the revised manuscript, we now used the Gag-mTagBFP2 proteins for finding specific correlations between lifetime or amplitude and Gag(-mTagBFP2) localization as well. There were negative or positive correlations between lifetime τ_1 or amplitude α_1 and mTagBFP2 intensity in Gag WT and Gag- Δ L expressing cells but not other mutants (except for a slight correlation between amplitude α_1 and mTagBFP2 intensity in EE mutant). These results support the idea that membrane curvature plays a role in the reorganization of lipid domains. We included these new findings in Fig. 6d-f.

In FRAP experiments, 1) the pinhole of the confocal microscope was fully opened, according to Snapp et al (2003), to collect as much fluorescence as possible during acquisition; and 2) the bleached area was too large (around 1.7 μ m in radius). These characteristics did not allow us to distinguish the area with and without Gag in the FRAP experiment, as shown in Fig. 4a, 4b, 4e, and 4f.

Snapp, E. L., Altan, N., and Lippincott-Schwartz, J. (2003) Measuring protein mobility by photobleaching GFP chimeras in living cells. *Curr Protoc Cell Biol* **Chapter 21**, Unit 21.1.

Moreover, one of the main advantages of such techniques is that they are combined with imaging. No images of such experiments are shown throughout the manuscript. I would encourage the authors to show images of these experiments.

We newly added representative images of cells labeled with EGFP-NT-Lys (Fig. 4a) and with EGFP-D4 (Fig. 4e) and fluorescence recoveries over time (Fig. 4b and 4f) for FRAP experiments, and representative images of EGFP-NT-Lys, Gag-mTagBFP2 derivatives, and lifetime τ_1 (Supplementary Fig. 11, Fig. 6d, and Supplementary Fig. 13) for FLIM-FRET experiments.

14. Figs. S3D to F are missing from the Supplementary Information file made available to me. Figure S4G is also missing.

We are sorry for this error. In the revised manuscript, the presence of all figures was double-checked.

15. The authors claim that “The FLIM diagram plot [for the Gag-WM mutant, which is oligomerisation incompetent] was similar to that without Gag expression”. However, it also seems similar to that with Gag expression (Fig. 4C). The amplitude values shown in Table 1 are indeed closer to those without Gag, however the lifetime (1.27 ns) is the same as in the Gag expressing cells. Further statistical treatment of these data would help clarify this point. While the plots showing the amplitude/lifetime values for each pixel are appreciated to understand the raw data, comparison between average values for each cell and independent experiment would show the experimental variability and establish the significance of the observed differences.

To perform a more statistically relevant comparison, we repeated the experiments independently three times. The value in each cell, the mean in each experiment, and the mean \pm SEM of the FRET lifetime and its amplitude for all experiments were plotted as in Fig. 5 and 6 of the revised version to show the experimental variability. A two-tailed paired t-test and a one-way ANOVA post-hoc Tukey test were applied to establish the significance of the observed differences. As described above, we moved the FLIM plots to the Supplementary Fig. 12.

16. The kernel density estimate function used to determine the number of populations in the FLIM data is not explained in the methods section. Have the authors validated that this method can distinguish four lifetime populations, as they do in Fig. 5 and Table 2 for the DA and WM samples (3 values for tau1 and 1 for tau2)? It also seems likely that the variability of the data just indicates that there is a broad range of FLIM values corresponding to a range of FRET distances between donor and acceptor, and not necessarily four distinct populations.

We added the explanation of the kernel density estimate function, which was used to smooth the probability distribution and find the population peaks, in the Supplementary Text (“Kernel density estimation used in flimDiagRam”). This function has been validated in our previous paper (Godet and Mely), where we showed that as many as three-lifetime peaks can be extracted from the FRET species described by the short-lived lifetime. As mentioned in the previous point, we have repeated the FRET measurements and their analysis. We found now that the FRET species distribute in only two populations. These two populations may correspond to different distances between donor and acceptor and/or different distributions of acceptors around the donor. This explanation has been added in the revised text (page 9, line 334-336).

Godet, J., and Mely, Y. Exploring protein-protein interactions with large differences in protein expression levels using FLIM-FRET. *Methods Appl. Fluoresc.* **8**, 014007 (2019).

The abovementioned claim “These data show for the first time the presence of heterogeneous complexes or domains that may differ by the arrangement and/or packing of SM and Chol molecules

using two different lipid probes.” Is solely based on this analysis, and as thus might be too speculative.

We agree with this comment and we thus tuned down the mentioned claim in the following way
“ The above data suggests the presence of heterogeneous complexes or domains that may differ by the arrangement and/or packing of SM and Chol molecules.” (page 14, line 519-521).

To further proof this, SMLM experiments studying the colocalisation between D4 and NT-Lys would be valuable.

We thank the reviewer for this comment. As suggested, we newly performed SMLM experiments for the colocalization between JF549-D4 and AF647-NT-Lys in the absence and presence of Gag expression. This result further supports our data that Gag expression induces Chol-rich and SM-rich domains to be in close proximity in FLIM-FRET. We included these results in Fig. 6g and 6h.

17. Which mounting medium was used for SMLM experiments?

To repeat experiments and answer the comment from reviewer#1 about AF647 contamination into mEos channel, we changed the mounting medium used in the former manuscript to a general buffer for STORM imaging (10% (w/v) glucose, 0.56 mg/mL glucose oxidase, 0.17 mg/mL catalase, and 100 mM beta-mercaptoethylamine). We added these details in “PALM/dSTORM” section in Methods (page 19, line 701-703).

18. Why is two-photon FLIM used instead of one-photon FLIM? Does it offer any advantages in this case? The complex photophysics of two-photon excitation might hinder the interpretation of the results.

Since its implementation, our FLIM set-up is equipped with a two-photon source that offers the advantage of providing a confocal excitation with a pulsed laser suited for FLIM experiments. Two-photon excitation provides less photobleaching than one-photon excitation, as only the focal volume is excited. Although two-photon and one-photon absorption mechanisms differ, the same S1 excited state is populated by the two excitation modes. As a result, the observed fluorescence lifetimes correspond to the same S1->S0 transition and are thus similar to the lifetimes reported in the literature (Volkmer et al., 2000).

Volkmer, A., V. Subramaniam, D.J. Birch, and T.M. Jovin. One- and two-photon excited fluorescence lifetimes and anisotropy decays of green fluorescent proteins. *Biophys. J.* **78**, 1589-1598 (2000).

19. The FLIM-FRET methods section misses some critical details. For example, what imaging medium was used? How long after transfection were the cells imaged? Is the imaging performed in the lower (coverslip) or upper plasma membrane? What is the pixel size? How long is the dwell time? Does the “binning off two” refer to two pixels?

We changed some conditions to repeat and add the experiment with the new Gag construct (Gag-EE) according to the referee’s suggestion. Therefore, we added all these details in “FLIM-FRET” in Methods section (page 20-21, line 730-792). To the above questions, 1) the imaging medium was PBS; 2) the cells were labeled with lipid probes 24 h after transfection with Gag plasmids, then fixed with 4% paraformaldehyde, and imaged next day; 3) the pixel size was around 230 nm; 4) the dwell time was 4 μ s/pixel; 5) the binning of two refers to 5 \times 5 pixels.

20. The Data and materials availability statement claims: “All data are available in the main text or the supplementary materials”. The raw data is not available in the figures, and since non-standard analysis methods are used, it would be valuable for the community if these data would be made available, even if it is upon request to the authors.

We have changed this to be consistent with the statement in the reporting summary.

Reviewer #3 (Remarks to the Author):

The authors present a study into the mechanism behind the different lipid compositions of the HIV envelope and the plasma membrane of the producer cell from which it is derived. A particular focus is given to the outer leaflet lipid sphingomyelin, which is enriched in virions. The authors recently developed a sphingomyelin-specific probe, a non-toxic mutant of lysenin, and this paper is a demonstration of the utility of this probe particularly with respect to super-resolution fluorescence imaging. The manuscript is well-written, follows a logical flow, and the conclusions are well-supported by the data. Key findings of the paper are that the HIV Gag protein (responsible for virus budding) colocalised with sphingomyelin clusters and causes their reorganisation or coalescence into larger clusters despite Gag accumulating on the inner leaflet, and sphingomyelin in the outer leaflet. FRAP experiments suggest that Gag increases the ‘immobile fraction’ of sphingomyelin, restricting its diffusion when associated with Gag.

D4 (a cholesterol-specific probe) is used alongside lysenin to show the coalescence of sphingomyelin- and cholesterol-rich domains at Gag sites. The authors suggest that membrane curvature plays a role in the coalescence using Gag mutants: P99A which does self-assemble but does not cause membrane curvature and does not bud, and Gag-deltaL which does curve the membrane but is unable to recruit ESCRT machinery also required for budding. It would have been compelling to see the effect of these

mutants on the previous experiments (colocalization, cluster size, and lateral diffusion) to exclude any changes in these properties due to mutation, leaving membrane curvature as the most-likely explanation for the sphingomyelin/cholesterol coalescence. It would also be useful to see (by EM) the nature of the curvature induced by these Gag mutants in these cells.

We thank the referee for this suggestion. Accordingly, we performed new super-resolution experiments to answer this comment. We notably performed PALM/STORM imaging of wild-type Gag, Gag-deltaL, -P99A, -EE (another curvature mutant), and -WM mutant expressing cells labeled with NT-Lys to see the colocalization between Gag and NT-Lys-labeled domains and cluster sizes of Gag and NT-Lys-labeled domains (Fig. 2, 3, and 7). Moreover, we recorded PALM/STORM images of a vector, wild-type Gag, or Gag-WM mutant-expressing cells labeled with D4 to see the colocalization between Gag and D4-labeled domains and cluster sizes of Gag and D4-labeled domains (Fig. 2 and 3). We additionally carried out FRAP experiments using D4 to see the lateral diffusion in the absence and presence of Gag (Fig. 4). All these data allowed us to link all results together and highlight the key role of membrane curvature on Gag reorganization of lipid domains in the plasma membrane. We also cited previous papers reporting the EM pictures of the curvatures formed by Gag mutants used in this study in the text (for Gag-deltaL, page 10-11, line 379-381; for Gag-P99A and Gag-EE, page 10, line 376-379).

Importantly, expression of Gag does not change the global lipid composition in the cell. This result is an important control, and should probably be stated up front in the results section, rather than being added at the end. Without this result the other results would not be interpretable. It should also be performed for Gag mutants.

As suggested, we examined the effect of Gag expression on the total lipid composition including SM and Chol for all Gag mutants and shifted the whole description at the beginning of the Results section (page 4, line 140-151, "Expression of Gag does not alter the gross lipid composition of HeLa cells"). The new data showing the absence of the impact of wild-type Gag and Gag mutants on lipid composition are gathered in Fig. 1 of the revised version.

One important aspect of Gag that has not been mentioned in this manuscript is that the protein is N-terminally myristoylated. The myristoylation has, in the past, been suggested to target Gag to certain lipid domains. This manuscript would suggest that curvature is the main feature driving sphingomyelin colocalization and restructuring. It would be interesting to see what effect a myristoylation-deficient Gag mutant has on colocalization and sphingomyelin/cholesterol coalescence.

It is well known that the myristoylation-deficient mutant of Gag is not recruited to the plasma membrane (Bryant and Ratner, 1990; Hermida-Matsumoto and Resh, 2000; Yu et al., 1995). Thus,

this Gag mutant is expected not to interact with sphingomyelin/cholesterol and impact the colocalization or the sphingomyelin/cholesterol domain reorganization across the plasma membrane.

Bryant, M., and L. Ratner. Myristoylation-dependent replication and assembly of human immunodeficiency virus 1. *Proc. Natl. Acad. Sci. U. S. A.* **87**, 523-527 (1990).

Hermida-Matsumoto, L., and M.D. Resh. Localization of human immunodeficiency virus type 1 Gag and Env at the plasma membrane by confocal imaging. *J. Virol.* **74**, 8670-8679 (2000).

Yu, G., F.S. Shen, S. Sturch, A. Aquino, R.I. Glazer, and R.L. Felsted. Regulation of HIV-1 gag protein subcellular targeting by protein kinase C. *J. Biol. Chem.* **270**, 4792-4796 (1995).

Minor comment: Fig 2B and S3F should be reworked for clarity. It isn't intuitively obvious the right graph is a zoom of the left, and it is not clear that the 50nm columns are truncated in the y-axis. Consider a split y-axis, or omitting the low value x-axis columns in the zoomed in data.

As suggested, we changed the data presentation in the revised manuscript and we drew graphs carefully following the reviewer's comment.

Reviewers' Comments:

Reviewer #1:

Remarks to the Author:

The authors have done a sizeable revision of their entire study and addressed all of my major and minor points sufficiently. I feel that this study should now be accepted for publication.

Reviewer #2:

Remarks to the Author:

The authors have addressed my initial comments satisfactorily.

This revised version is of high quality and relevant to the field.

I support the publication of this work after the following comments are addressed:

-Validation of fluorescent Gag is done with the EGFP version, but Gag-EGFP is not used in the microscopy experiments. I suggest performing the validation experiments in Sup. Figs. 2 and 3 with Gag-mBFP2tag2 and Gag-mEos4b.

-Colocalisation (MOC) values of Gag-mEos4b WT and AF647-NT-Lys (Fig 2e) and AF647N-D4 (Fig. 2j) are high (around 50-60%), however microscopy images (Figs. 2c and h) do not seem show high colocalisation. Why is this?

-In line with the previous comment. Does the Voronoi colocalisation approach consider as colocalised signals that do not overlap with each other but are instead close ("a given data point is a locus of all points in space closest to this data point"). If so, is it artificially reducing resolution to increase colocalisation?

The authors should explain the advantage of using Voronoi diagrams for colocalisation.

-Fig. 6c, the sample names are not aligned to the data points.

-Fig 6e and f, the quartile grouping is strange. I suggest scatter plots instead, where x is the mTagBFP2 intensity and y tau1 and alpha1, respectively.

Reviewer #3:

Remarks to the Author:

The authors have satisfactorily addressed all my comments from the previous review.

A point-by-point response to REVIEWERS' COMMENTS on NCOMMS-21-34466A

Following the policy of the journal, we added the methods and references in supplementary information to the main text.

Reviewer #1 (Remarks to the Author):

The authors have done a sizeable revision of their entire study and addressed all of my major and minor points sufficiently. I feel that this study should now be accepted for publication.

We thank the reviewer for reviewing our revised manuscript and for the positive feedback.

Reviewer #2 (Remarks to the Author):

The authors have addressed my initial comments satisfactorily.

This revised version is of high quality and relevant to the field.

I support the publication of this work after the following comments are addressed:

We thank the reviewer for reviewing our revised manuscript and for the positive comments.

-Validation of fluorescent Gag is done with the EGFP version, but Gag-EGFP is not used in the microscopy experiments. I suggest performing the validation experiments in Sup. Figs. 2 and 3 with Gag-mBFPTag2 and Gag-mEos4b.

In EGFP, a chromophore is surrounded by the characteristic 11-stranded beta-barrel, which is common among fluorescent proteins. Both mEos4b (protein data bank ID (PDB ID): 6GOY, De Zitter et al (2019) Nat Methods) and mTagBFP (PDB ID: 3M24, Subach et al (2010) Chem Biol) share the same barrel structure and size (mEos4b, 25.9 kDa; mTagBFP2, 26.7 kDa) as EGFP (PDB ID: 2Y0G, 26.9 kDa) and differ only by their chromophores inside (Reviewer only Fig. 3). Furthermore, both mEos4b and mTagBFP were engineered to be monomeric (Paez-Segala et al (2015) Nat Methods; Subach et al (2008) Chem Biol) and were inserted into the same position of Gag as Gag-EGFP. Thus, we presume that Gag fused to both proteins works in the same manner as the Gag-EGFP protein.

Nevertheless, we performed the validation experiments of Gag-mEos4b and Gag-mTagBFP2 using VLP release as suggested by the reviewer. In Supplementary Fig. 4, transfection with a 3:1 mixture of non-tagged Gag and Gag-mEos4b or Gag-mTagBFP2 showed that VLP was produced with the same efficiency as Gag-EGFP. This result further strongly confirms that Gag-mEos4b and Gag-mTagBFP2 work in the same as the Gag-EGFP.

De Zitter E, Thédié D, Mönkemöller V, Hugelier S, Beaudouin J, Adam V, Byrdin M, Van Meervelt L, Dedeker P, Bourgeois D. Mechanistic investigation of mEos4b reveals a strategy to reduce track interruptions in sptPALM. Nat. Methods **16**, 707-710 (2019).

Subach OM, Malashkevich VN, Zencheck WD, Morozova KS, Piatkevich KD, Almo SC, Verkhusha VV. Structural characterization of acylimine-containing blue and red chromophores in mTagBFP and TagRFP fluorescent proteins. *Chem. Biol.* **17**, 333-341 (2010).

Paez-Segala MG, Sun MG, Shtengel G, Viswanathan S, Baird MA, Macklin JJ, Patel R, Allen JR, Howe ES, Piszczek G, Hess HF, Davidson MW, Wang Y, Looger LL. *Nat. Methods* **12**, 215-218 (2015).

Subach OM, Gundorov IS, Yoshimura M, Subach FV, Zhang J, Grünwald D, Souslova EA, Chudakov DM, Verkhusha VV. Conversion of red fluorescent protein into a bright blue probe. *Chem. Biol.* **15**, 1116-1124 (2008).

- Colocalisation (MOC) values of Gag-mEos4b WT and AF647-NT-Lys (Fig 2e) and AF647N-D4 (Fig. 2j) are high (around 50-60%), however microscopy images (Figs. 2c and h) do not seem show high colocalisation. Why is this?

The reason of this apparent discrepancy is that Fig. 2c or h is only a part of the whole cell image. To get a better idea on the difference in colocalization between AF647-NT-Lys or AF647-D4 and Gag-mEos4b, Fig. 2b and 2g would be more appropriate. However, even in this case, it remains extremely difficult to compare the super-resolution images by eyes since the localizations that are not in dense clusters are hardly recognizable by eyes, but meet the density criteria (see below) and are counted in the colocalization analysis.

-In line with the previous comment. Does the Voronoi colocalisation approach consider as colocalised signals that do not overlap with each other but are instead close ("a given data point is a locus of all points in space closest to this data point"). If so, is it artificially reducing resolution to increase colocalisation?

The authors should explain the advantage of using Voronoi diagrams for colocalisation.

The Voronoi colocalization approach used in the program (PoCA and/or Coloc-Tesseler (Levet et al (2019) *Nat Commun*)) computes a normalized density that is automatically determined from the image to be analyzed and used this normalized density as a threshold. In this sense, the analysis procedure is free from any bias from the user. Therefore, here we focus on the advantage of Coloc-Tesseler used in this study. 1) The colocalization analysis by this program is autonomous due to the normalized density, which as mentioned above, ensures consistent analysis without any assumptions when analysis is needed for multiple images. 2) Compared to CBC (coordinate-based colocalization) used in the colocalization analysis of our previous manuscript, the results (MOC) obtained by this method are more intuitive and easier to understand because it can be sometimes difficult to explain negative values associated with CBC analysis. As the reviewer pointed out, a diagram of one channel localization can overlap with another channel diagram where the localization is not in the center. This point is a possible artifact inherent to Voronoi methods. However, to avoid this edge artifact, the program integrates an edge correction parameter (as shown in

Supplementary Fig. 3 in Levet et al (2019) Nat Commun) and removes overestimated colocalizations.

Levet F, Julien G, Galland R, Butler C, Beghin A, Chazeau A, Hoess P, Ries J, Giannone G, Sibarita JB. A tessellation-based colocalization analysis approach for single-molecule localization microscopy. Nat. Commun. 10, 2379 (2019).

-Fig. 6c, the sample names are not aligned to the data points.

We thank the referee for pointing this out. We corrected the alignment of sample names against the data points.

-Fig 6e and f, the quartile grouping is strange. I suggest scatter plots instead, where x is the mTagBFP2 intensity and y tau1 and alpha1, respectively.

In Fig. 6e and f, we measured tau1 and alpha 1 values in single cells. In such experiments, especially in areas with low intensities of (Gag-)mTagBFP2, tau1 and alpha1 values were highly variable. This makes the correlation analysis difficult to show in scatter plots where x is the mTagBFP2 intensity and y tau1 and alpha1, respectively. We thus used the quartile grouping and averaging at each quartile.

Reviewer #3 (Remarks to the Author):

The authors have satisfactorily addressed all my comments from the previous review.

We thank the reviewer for reviewing our revised manuscript and for the comment.